# A comprehensive overview of genomic imprinting in breast and its deregulation in cancer

Tine Goovaerts[1], Sandra Steyaert[1], Chari A. Vandenbussche[1], Jeroen Galle[1], Olivier Thas[1,2,3], Wim Van Criekinge[1,2,3] & Tim De Meyer[1,2,3]

Genomic imprinting plays an important role in growth and development. Loss of imprinting (LOI) has been found in cancer, yet systematic studies are impeded by data-analytical challenges. We developed a methodology to detect monoallelically expressed loci without requiring genotyping data, and applied it on The Cancer Genome Atlas (TCGA, discovery) and Genotype-Tissue expression project (GTEx, validation) breast tissue RNA-seq data. Here, we report the identification of 30 putatively imprinted genes in breast. In breast cancer (TCGA), *HM13* is featured by LOI and expression upregulation, which is linked to DNA demethylation. Other imprinted genes typically demonstrate lower expression in cancer, often associated with copy number variation and aberrant DNA methylation. Downregulation in cancer frequently leads to higher relative expression of the (imperfectly) silenced allele, yet this is not considered canonical LOI given the lack of (absolute) re-expression. In summary, our novel methodology highlights the massive deregulation of imprinting in breast cancer.

[1] Department Data Analysis and Mathematical Modelling, Ghent University, Coupure Links 653, 9000 Ghent, Belgium. [2] Bioinformatics Institute Ghent - from Nucleotides to Networks (BIG N2N), Ghent University, Technologiepark 927, 9052 Ghent, Belgium. [3] Cancer Research Institute Ghent (CRIG), Ghent University, Corneel Heymanslaan 10, 9000 Ghent, Belgium. These authors contributed equally: Tine Goovaerts, Sandra Steyaert. Correspondence and requests for materials should be addressed to T.D.M. (email: tim.demeyer@ugent.be)

Breast cancer is the most common type of cancer in women[1]. It is a very heterogeneous disease with major differences in incidence, clinical outcome, prognosis and response to therapy[2,3]. Gene expression profiling led to the division of breast cancer in five different molecular subtypes: Luminal A, Luminal B, HER2-enriched, Basal-like and Normal-like[2,4]. These subtypes differ amongst others in expression of the oestrogen receptor, progesterone receptor, human epidermal growth factor receptor 2 (HER2) and in histological grade[3].

An early occurring aberration in cancer is loss of imprinting (LOI)[5]. Imprinting refers to the monoallelic expression of genes in a parent-of-origin-specific manner. In diploid eukaryotic organisms, the maternal and paternal copies of most genes are expressed at similar levels. For imprinted genes, however, only a single allele is transcriptionally active[6–8]. Imprinting patterns may vary between tissues[9]. Imprinted genes are mostly clustered and regulated by imprinting control regions, which are typically under DNA methylation control, though also H3K27me3 was demonstrated to be involved[10,11]. Imprinted genes play an important role in development and placental biology[12]. Furthermore, as dosage of imprinted genes is crucial, disruption of imprinting can result in a number of human imprinting syndromes and may predispose to cancer by promoting tumourigenic or suppressing antitumour mechanisms[13–16]. Some well-known diseases are Angelman Syndrome (functional loss of the maternal, active allele of *UBE3A*), Prader–Willi Syndrome (loss of the paternal, active allele of *SNRPN*) and Beckwith–Wiedemann Syndrome (LOI on chromosome 11)[14,17].

LOI results in biallelic expression due to activation of the silent allele. Indeed, experiments in mice demonstrated that demethylation at imprinted genes leading to LOI made cells susceptible to cellular transformation and tumourigenesis[18]. For instance, aberrant biallelic expression of the imprinted *IGF2* locus is thought to promote tumourigenesis by inhibiting apoptosis in colorectal cancer[19] and to lead to over-proliferation defects in lung, colon and ovarian cancer[20]. LOI of other imprinted genes, such as *H19*, *PEG3*, *MEST* and *PLAGL1*, was also discovered in varying cancers[21].

However, several studies suggest a far more complicated story, where LOI is associated with silencing of the normally active allele[5]. Indeed, recent studies identified major expression downregulation of reportedly imprinted genes in cancer[22,23]. Moreover, in oesophageal cancer, LOI of *IGF2* was specifically associated with expression downregulation, and improved survival[24]. Also in prostate cancer, no increased expression was found for *IGF2* despite LOI[25]. Notwithstanding the major relevance of LOI in cancer, this fragmentary evidence demonstrates that the current paradigm of the role of LOI in cancer (i.e. growth & tumour promoting expression) requires additional evaluation. A recent study by Ribaraska et al. found downregulation of several imprinted genes in prostate cancer, but stable DNA methylation[23]. These results suggest the existence of an imprinted gene network in which these genes are co-regulated, as was also observed in mice[26]. Recently, also copy number variation (CNV) was identified as an important cause of imprinting deregulation in cancer[27].

Systematic analyses of LOI are still lacking. Indeed, although monoallelic expression is a well-investigated topic, only few regions are well-characterised in humans, and only a single study thoroughly evaluated tissue-specific imprinting patterns[9]. To date, the impact of aberrant monoallelic expression on cancer has typically been studied at single imprinted loci[18]. Moreover, the practical applicability of existing high-throughput methods is greatly hampered by the necessity for genotyping next to (typically) RNA-seq data. Thus, there is a need to systematically profile (i) monoallelically expressed/imprinted loci and (ii) their deregulation (LOI) in cancer, preferably solely based on RNA-seq data.

Here, such a methodology is presented and—given indications for massive differential expression of imprinted genes in this tumour[22]—applied on breast control (TCGA[28], GTEx[29]) and cancer data (TCGA), leading to the identification of 30 putatively imprinted genes in breast of which 8 are featured by increased biallelic expression in at least one breast cancer subtype. Comparison with whole-exome sequencing (WES) data demonstrate that (i) the RNA-seq-based results are generally reliable, and (ii) that avoiding the use of WES data leads to a far higher genome-wide character. Intriguingly, the results indicate that the increased frequency of biallelic expression is far more often associated with lower expression than higher expression of the corresponding locus, though exceptions exist (e.g. *HM13*). Therefore, this study demonstrates that deregulation of imprinting is an important feature in (breast) cancer but is not automatically associated with canonical LOI. Furthermore, these results underline the efficacy of the proposed strategy for the identification of imprinted regions and their deregulation.

## Results

**Detection of imprinting in healthy breast tissue.** First, a methodology was developed and applied to screen for imprinted loci in RNA-seq data, using single nucleotide polymorphisms (SNPs) to discriminate between alleles. Contrasting previous genome-wide methods, no DNA genotyping data is required, as we solely rely on genotyping of RNA-seq data. The basic rationale is that in case of 100% imprinting, no heterozygous samples can be found in RNA-seq data, as they perfectly resemble homozygous samples (only a single allele is expressed) (see Methods section and Steyaert et al.[30]). The novel imprinting model describes the data for each SNP as a mixture of homozygous and heterozygous samples, more specifically as a mixture of genotype-dependent binomial distributions, with weights derived from Hardy–Weinberg equilibrium. The model allows for sequencing errors and partial imprinting. Indeed, one parameter describes the degree of imprinting, and it can be evaluated whether its estimate is significantly higher than 0 using a likelihood ratio test.

Upon application on 113 TCGA breast control samples, 127 SNPs were considered to be possibly imprinted (false discovery rate (FDR) ≤ 0.05), and dbSNP annotation was found for 125 SNPs. The 125 possibly imprinted SNPs corresponded with approximately 35 genes, the majority already known to be imprinted (Supplementary Note 1–3). Note that annotation of the SNPs to specific genes was often difficult as many overlapping genes were found (Supplementary Note 1). For example, *MCTS2P* is a retrogene copy and located in *HM13*, making it uncertain in which gene the detected SNP was located[31]. Similarly, for *MTRNR2L1*, imprinted SNPs with Ensembl annotation for this locus were found upstream of the gene, making correct annotation uncertain. Upon validation in GTEx healthy breast data, 121 SNPs corresponding to 30 genes remained (96.8% SNP validation rate). Table 1 lists the identified SNP loci and corresponding genes. As examples, the resulting mixture distributions of *IGF2* and *SNRPN* are shown in Fig. 1a, b, respectively. The distributions show that these loci are clearly depleted of samples featured by biallelic expression. In Fig. 1c a non-imprinted SNP with a distinct heterozygous peak is shown. Moreover, Fig. 1a indicates that *IGF2* imprinting is only partial (97% imprinted), underscoring the suitability of the flexible distribution model used here. Similar figures for the other genes can be found in Supplementary Figs. 1–11. In general, imprinting could be verified using TCGA WES data, yet for many SNPs this was technically infeasible due to too low coverages—supporting

**Table 1 Genes featured by monoallelic expression/imprinting in breast tissue[a]**

| Gene symbol | SNPs | Gene symbol | SNPs | Gene symbol | SNPs |
|---|---|---|---|---|---|
| MTCO1P12[c] | rs112232512 (1.42E-75) | MEG3 | rs3087918 (5.47E-27) | SNHG14 | rs4344720 (7.48E-20) |
| LINC01139 | rs61746209 (6.14E-29) | MEG3 | rs3087917 (1.64E-30) | SNHG14 | rs3863396 (2.68E-19) |
| ZDBF2 | rs7582864 (1.02E-21) | MEG3 | rs3742391 (2.13E-19) | SNHG14 | rs62002013 (3.83E-12) |
| ZDBF2 | rs3732084 (6.54E-26) | MEG3 | rs12897172 (2.31E-17) | SNHG14 | rs2356294 (4.96E-24) |
| ZDBF2 | rs1975597 (7.97E-30) | MEG3 | rs1884540 (1.78E-26) | SNHG14 | rs1043164 (6.41E-12) |
| ZDBF2 | rs1448902 (2.02E-17) | MEG3 | rs2400941 (1.64E-29) | SNHG14 | rs691 (6.11E-33) |
| ZDBF2 | rs4673350 (2.56E-23) | MEG3 | rs77658190 (2.45E-17) | SNHG14 | rs13526 (3.74E-29) |
| PAX8-AS1 | rs7585510 (0.0068) | MEG3 | rs10132552 (4.26E-17) | PLIN1[b] | rs4578621 (3.29E-06) |
| PTX3[b] | rs73158510 (5.44E-07) | MEG3 | rs3194464 (1.43E-23) | ZNF597 | rs37822 (1.47E-18) |
| NAP1L5 | rs8605 (3.77E-22) | MEG3 | rs11160606 (1.53E-23) | ZNF597 | rs37823 (3.08E-18) |
| NAP1L5 | rs710834 (9.10E-21) | MEG3 | rs1950628 (1.01E-26) | ZNF597 | rs11639510 (3.42E-25) |
| ZNF300P1 | rs17800987 (4.87E-11) | MEG3 | rs1053900 (4.91E-34) | ZNF597 | rs37824 (3.21E-18) |
| PLAGL1 | rs2328535 (3.09E-10) | MEG3 | rs1054000 (7.58E-27) | ZNF597 | rs12737 (1.01E-21) |
| PLAGL1 | rs9373409 (1.40E-27) | MEG3 | rs8013873 (6.63E-27) | USP32P2[b] | rs141915702 (0.0093) |
| PLAGL1 | rs73006222 (2.41E-15) | MEG3 | rs11859 (2.17E-25) | MTRNR2L1[c] | rs3931649 (3.16E-28) |
| PLAGL1 | rs17615967 (1.76E-14) | MEG3 | rs74080162 (1.15E-24) | MTRNR2L1[c] | rs113014658 (6.82E-09) |
| PLAGL1 | rs77203559 (3.76E-15) | MEG3 | rs4378559 (8.76E-25) | MTRNR2L1[c] | rs113626706 (5.74E-12) |
| PLAGL1 | rs9321953 (1.48E-21) | MEG3 | rs12890215 (8.74E-25) | MTRNR2L1[c] | rs3931650 (7.67E-29) |
| LOC100294145 | rs241407 (2.71E-07) | MEG3 | rs55996894 (5.63E-21) | ZNF331 | rs113983639 (1.99E-13) |
| PEG10 | rs35237090 (1.69E-17) | MEG3 | rs3742390 (6.24E-31) | ZNF331 | rs8110350 (4.12E-106) |
| PEG10 | rs13073 (3.35E-13) | MEG3 | rs4906022 (8.42E-204) | ZNF331 | rs8110538 (1.74E-109) |
| PEG10 | rs7810469 (6.18E-29) | SNRPN | rs2554426 (9.39E-16) | ZNF331 | rs8109631 (8.52E-175) |
| MEST | rs10863 ( < detection limit) | SNRPN | rs705 (1.81E-71) | PEG3 | rs4801386 (0.00011) |
| HOTAIRM1 | rs706018 (5.67E-36) | SNHG14 | rs2732028 (3.33E-15) | PEG3 | rs1558355 (0.00027) |
| H19 | rs2075745 (6.63E-34) | SNHG14 | rs74335291 (5.12E-24) | PEG3 | rs723082 (1.62E-31) |
| H19 | rs2075744 (3.21E-27) | SNHG14 | rs2732029 (6.71E-28) | PEG3 | rs3143 (7.86E-06) |
| H19 | rs2839698 (3.18E-57) | SNHG14 | rs765438 (1.96E-28) | PEG3 | rs1055359 (9.07E-20) |
| H19 | rs2067051 (1.94E-57) | SNHG14 | rs2732030 (9.43E-12) | PEG3 | rs11666110 (8.06E-28) |
| H19 | rs2839701 (3.29E-61) | SNHG14 | rs10451029 (6.14E-22) | PEG3 | rs1860565 (1.02E-21) |
| H19 | rs2839704 (1.40E-44) | SNHG14 | rs2052723 (1.14E-25) | PEG3 | rs33931963 (6.73E-28) |
| H19 | rs2839702 (2.94E-50) | SNHG14 | rs2554419 (7.96E-31) | HM13 | rs6058058 (1.33E-10) |
| H19 | rs2839703 (1.10E-46) | SNHG14 | rs719704 (3.32E-22) | HM13 | rs6059869 (4.04E-14) |
| H19 | rs10840159 (2.28E-50) | SNHG14 | rs34316840 (1.13E-25) | HM13 | rs6059873 (1.29E-19) |
| H19 | rs3741219 (1.01E-107) | SNHG14 | rs2732031 (1.23E-26) | HM13 | rs6059874 (4.88E-14) |
| RP11-109L13.1[b] | rs201284359 (4.97E-10) | PWAR6 | rs2732041 (1.60E-12) | HM13 (MCTS2P) | rs1115713 (7.94E-09) |
| IGF2 | rs7873 (1.08E-268) | PWAR6 | rs2732043 (1.78E-13) | GNAS/GNAS-AS1 | rs1800900 (5.02E-16) |
| IGF2 | rs2585 (< detection limit) | PWAR6 | rs2732044 (1.42E-26) | BCR | rs550197 (7.93E-50) |
| DLK1 | rs1802710 (2.68E-28) | PWAR6 | rs1030389 (1.68E-28) | CPHL1P | rs12497062 (0.00027) |
| MEG3 | rs78793760 (4.16E-21) | PWAR6 | rs62001981 (1.41E-26) | ATP8A1 | rs11940243 (3.21E-08) |
| MEG3 | rs35458454 (5.16E-21) | PWAR6 | rs62001982 (1.09E-19) | GLIPR1/KRR1 | rs1056905 (1.20E-06) |
| MEG3 | rs35431412 (1.62E-29) | PWAR6 | rs1045935 (6.76E-29) | TPSB2 | rs77309587 (1.52E-08) |
| MEG3 | rs10147988 (1.07E-26) | SNHG14 | rs11637436 (1.44E-11) |  |  |

[a]The columns show the gene symbol (Gene symbol) and the ID of significantly imprinted SNP locus falling in these genes (SNPs) with its LRT FDR-adjusted p-value between brackets. Genes which are known to be imprinted or for which there is prior evidence of imprinting are marked in bold. Genes eliminated from our set of imprinted genes upon unsuccessful validation in GTEx are underlined.
[b]Candidate imprinted genes due to limited coverage/allele frequency in GTEx or inconsistent results per exon
[c]Candidate imprinted pseudogenes, involvement original mitochondrial genes could not be fully excluded (Supplementary Note 3e)

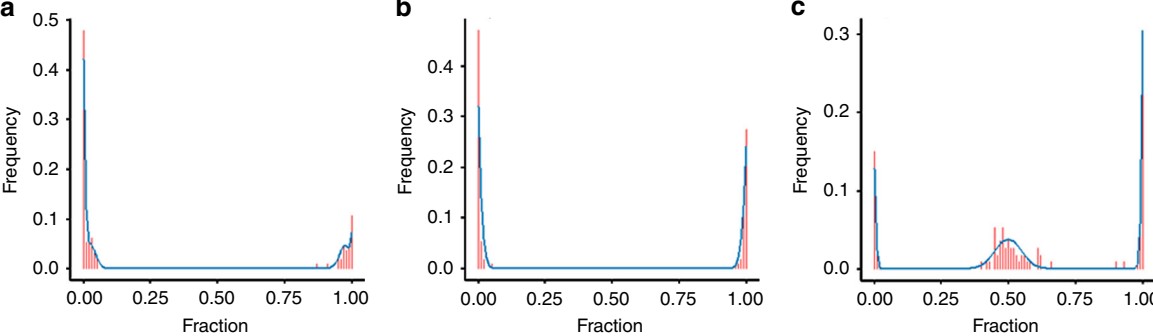

**Fig. 1** Mixture distributions of (non-)imprinted SNPs. Observed (red) and modelled (blue) fraction of alternative alleles for two significantly imprinted SNP positions, i.e. **a** IGF2 (rs2585, adj. p-value < detection limit (LRT), $\hat{\imath} = 0.97$) **b** SNRPN (rs705, adj. p-value = 1.81E-71 (LRT), $\hat{\imath} = 0.99$), and a non-imprinted SNP, i.e. **c** CHMP6 (rs1128687, $\hat{\imath} = 0$)

the value of the genotyping-free approach introduced here (Supplementary Note 3c).

Next to imprinting, the model detects random monoallelic expression, as this can only be discriminated from imprinting by means of family data, unavailable in TCGA. As a consequence, several *HLA*, *HLA-DR* and *HLA-DQ* genes—important players in immune reactions known to be regulated by random monoallelic expression[9]—were also picked up, but were excluded from further analyses. The remaining genes (Table 1) represent the final list of putatively imprinted loci in normal human breast tissue further analysed throughout this manuscript.

As typically only few putatively imprinted SNPs per gene were found, we evaluated the other SNPs in identified genes. Often SNPs were located in intronic regions or 5′-UTRs with too low coverage for accurate detection of imprinting. Other undetected SNPs in exons or 3'-UTRs were typically featured by a high sequencing error rate, low minor allele frequency or inferior goodness-of-fit to the model (Supplementary Data 1–2). As a result, most of these missed SNPs were filtered out prior to application of the imprinting likelihood ratio test, or exhibited non-significant results due to aforementioned problems. In Supplementary Data 1 all SNPs in *HM13* are shown, which clearly demonstrates that indeed mostly intronic SNPs with low coverages are missed. However, some genes showed evidence (*ZNF300P1*, *LOC100294145*, *ZNF331* and *GNAS-AS1/GNAS*) or indications (*HOTAIRM1*) of transcript-specific imprinting with more complex mixture distributions and consistent differences between exons (Supplementary Figs. 16–17 and Supplementary Data 2). On the other hand, for *PTX3*, *RP11-109L13*.1, *PLIN1* and *USP32P*, inconsistent imprinting patterns could not be readily explained by gene isoforms, and these loci are denoted as candidate imprinted (Supplementary Fig. 15 and Supplementary Data 2). Known possibly imprinted genes were clearly not imprinted in breast tissue independent of overlapping transcripts (Supplementary Figs. 18–19). For possible pseudogenes, we evaluated misalignment as possible explanation. For *MTRNR2L1* and *MTCO1P12*, this additional curation step did not lead to unequivocal results, leading us to also designate these loci as candidate imprinted (Table 1, Supplementary Note 3e).

As an additional validation step, we examined whether DNA methylation levels were skewed towards putative monoallelic methylation for the imprinted loci. For the 18 genes represented in the TCGA Infinium HumanMethylation data, at least one putative monoallelically-methylated probe was found (median methylation in between 33.3 and 66.7%) (Supplementary Data 3). Moreover, 46% of all probes for these loci were found to be featured by putative monoallelic methylation, a major enrichment compared to expected (all probes, 14%, $p = 1.84E-175$, $\chi^2$ test). However, monoallelic methylation is only a crude proxy for the presence of imprinting associated differentially methylated regions (DMR). We therefore analysed methylation in known DMR locations, provided in Table 1 of Court et al.[32]. As summarised in Supplementary Data 3, DMRs were found for all identified imprinted loci, except for *PTX3*, *PLIN1* and *BCR*, which may be regulated in a different manner. Typically, each of these DMRs featured multiple putatively monoallelically-methylated probes in the control samples, and often even largely coincided with putatively monoallelically-methylated regions (e.g. *PEG10*, *PLAGL1* and *HM13*). For *ZDBF2* and *DLK1*, the DMR methylation status could not be evaluated since no probes were present.

**Differential imprinting in breast cancer**. To examine possible deregulation of imprinting in breast cancer, alterations in allelic expression patterns of imprinted genes were investigated.

Differential imprinting (DI) is defined here as different relative expression of both alleles in cancer vs controls. We reserve the term LOI solely for DI caused by re-expression of the silenced allele (i.e. absolute higher expression). We determine biallelic expression per sample by the allelic ratio (allele count with lowest expression/allele count with highest expression), which varies from 0 (perfect monoallelic expression) to 1 (perfect biallelic expression) and is independent of expression level differences in cancer. For these analyses, it should be noted that power is compromised by the fact that DI cannot be observed in homozygous individuals, which constitutes the majority for each locus. We try to compensate this by allowing more false positives in the results (FDR of 10%), but also by enriching for likely heterozygous samples. As samples with a high allelic ratio represent heterozygotes, looking at the $2P_A P_T$ highest fraction of samples (cf. Hardy–Weinberg theorem) allows us to only take the most likely heterozygotes into account. DI in cancer for a specific SNP was thus defined as a significant difference of the allelic ratios between putative heterozygous cancer and control samples (see Methods, section Detection of differential imprinting). Though only technically feasible for a limited set of the data, this allelic ratio was verified to be a good measure for DI using WES data (Supplementary Fig. 14).

When considering the full set of 506 breast cancer samples, four SNP loci with significant DI, i.e. higher relative expression of the silenced allele, could be identified (FDR ≤ 0.1; Table 2). These SNPs correspond to three genes, namely *MEST*, *H19* and *HM13*. Figure 2 shows the mixture distributions of these genes for both control and tumour samples. The plots demonstrate that for *MEST* (and to a minor extent for *HM13*) also some control samples are featured by residual biallelic expression. This may suggest that this locus is less stringently imprinted and further lost its imprinting signature in breast cancer. Alternatively, this may be due to the fact that a mixture of imprinted and non-imprinted transcripts is present, with a shift towards expression of the latter in cancer.

Subsequently, the different breast cancer subtypes, namely basal-like (BL), HER2-enriched (HER2), luminal A (LumA) and luminal B (LumB), were analysed individually for DI compared to normal samples (Table 2, significant results are shown in bold). The normal-like (NL) subtype was not considered due to a too low number of samples. Compared to the results for all tumour samples, significant DI of *MEST* was detected at the same SNP position in all subtypes except for BL, whereas *HM13* and *H19* were significantly deregulated in all but LumA. In summary, most deregulation was found in HER2 and BL. The deregulated loci corresponded to eight genes, i.e. *ZDBF2*, *PEG10*, *MEST*, *H19*, *IGF2*, *MEG3*, *ZNF331* and *HM13*. For each of these genes, the frequency of samples featuring biallelic expression in cancer was estimated, going up to 100% for *MEST* in LumB and HER2 (Supplementary Table 5). For LumB, DI was found in *MEST*, *H19*, *HM13*, *IGF2*, *ZNF331* and *PEG10*, whereas for LumA only one DI locus was identified (Fig. 3). Distributions of the other loci featured by DI in cancer subtypes are displayed in Supplementary Figs. 20–23. Particularly the *H19/IGF2* locus showed striking evidence for DI in HER2 and BL samples: all 12 SNPs were found to be deregulated in BL, whereas for HER2 10 SNPs were DI. DI was not associated with survival, except for two SNPs in *ZDBF2*, nor with age (Fig. 4, Supplementary Note 4c).

Though most often the case, SNPs in the same imprinted gene did not always show consistent (in)significant results (Supplementary Data 4). This can typically be attributed to technical/power associated causes, such as lower coverage, though also transcript-specific effects may be present. For example, for *ZNF331*, two DI SNPs were found in the 3'-UTR, and one non-DI SNP in the first exon. Again, verification of DI by comparison

**Table 2 SNPs with significant differential imprinting in control samples versus breast cancer and the different subtypes[a]**

| SNP | Gene | Tumour | | | HER2-enriched | | | Basal-like | | | Luminal A | | | Luminal B | | |
|---|---|---|---|---|---|---|---|---|---|---|---|---|---|---|---|---|
| | | DI_p | logFC | DE_p | DI_p | logFC | DE_p | DI_p | logFC | DE_p | DI_p | logFC | DE_p | DI_p | logFC | DE_p |
| rs1053900 | MEG3 | 0.49 | −2.4 | **6.6E-45** | **0.07** | −2.2 | **5.6E-18** | 1.00 | −3.0 | **1.8E-37** | 0.90 | −2.2 | **1.6E-29** | 0.46 | −2.7 | **1.1E-35** |
| rs4378559 | MEG3 | 0.49 | −1.7 | **2.8E-27** | **0.04** | −1.0 | **2.0E-08** | 1.00 | −2.4 | **6.8E-26** | 0.92 | −1.4 | **1.6E-16** | 0.46 | −2.2 | **2.2E-26** |
| rs12890215 | MEG3 | 0.47 | −1.7 | **5.1E-29** | **0.10** | −1.0 | **5.7E-09** | 0.76 | −2.3 | **8.0E-27** | 0.87 | −1.4 | **9.1E-18** | 0.44 | −2.2 | **6.7E-28** |
| rs10863 | MEST | **0.02** | −1.6 | **1.3E-31** | **0.00** | −1.7 | **3.0E-13** | 0.70 | −0.9 | **6.5E-15** | **0.03** | −1.9 | **5.4E-31** | **0.00** | −1.7 | **2.7E-19** |
| rs3741219 | H19 | 0.17 | −0.9 | **1.4E-02** | **0.00** | −1.1 | **1.4E-03** | **0.03** | −0.8 | **4.5E-04** | 0.77 | −0.7 | 3.5E-01 | 0.38 | −1.3 | **7.8E-04** |
| rs2839704 | H19 | **0.07** | −0.9 | **2.1E-02** | **0.01** | −1.2 | **8.1E-04** | **0.02** | −0.8 | **9.7E-04** | 0.40 | −0.7 | 4.5E-01 | **0.03** | −1.2 | **8.3E-04** |
| rs2839703 | H19 | **0.08** | −0.9 | **2.7E-02** | **0.01** | −1.3 | **9.8E-04** | **0.02** | −0.8 | **1.4E-03** | 0.56 | −0.8 | 4.7E-01 | **0.03** | −1.3 | **9.8E-04** |
| rs10840159 | H19 | 0.17 | −1.0 | **3.4E-02** | **0.01** | −1.4 | **2.3E-03** | **0.02** | −0.9 | **1.3E-03** | 0.85 | −0.8 | 4.8E-01 | 0.27 | −1.3 | **1.3E-03** |
| rs2839702 | H19 | 0.17 | −0.7 | **3.3E-02** | **0.01** | −1.0 | **7.0E-03** | **0.02** | −0.5 | **4.9E-04** | 0.65 | −0.5 | 4.1E-01 | 0.16 | −1.0 | **4.2E-03** |
| rs2839701 | H19 | 0.17 | −0.6 | 6.4E-02 | **0.01** | −0.9 | **9.5E-03** | **0.03** | −0.5 | **1.1E-03** | 0.76 | −0.5 | 4.9E-01 | 0.22 | −0.9 | **8.7E-03** |
| rs2067051 | H19 | 0.19 | −0.3 | 4.8E-01 | **0.05** | −0.6 | 4.8E-01 | **0.02** | −0.2 | 1.1E-01 | 0.91 | 0.0 | **4.7E-03** | 0.20 | −0.6 | 1.9E-01 |
| rs2075745 | H19 | 0.17 | −0.4 | 8.6E-01 | **0.04** | −0.8 | 3.3E-01 | **0.02** | −0.4 | 6.1E-02 | 0.78 | −0.2 | **2.6E-02** | 0.16 | −0.7 | 1.4E-01 |
| rs2075744 | H19 | 0.27 | −0.2 | 4.1E-01 | 0.17 | −0.6 | 2.8E-01 | **0.02** | −0.3 | 1.9E-01 | 0.92 | 0.0 | **1.7E-02** | 0.57 | −0.4 | 3.6E-01 |
| rs2839698 | H19 | 0.23 | −0.3 | 4.5E-01 | **0.08** | −0.7 | 3.8E-01 | **0.02** | −0.4 | 1.4E-01 | 0.90 | −0.1 | **1.4E-02** | 0.38 | −0.5 | 3.7E-01 |
| rs7582864 | ZDBF2 | 0.49 | −1.7 | **2.7E-25** | 0.75 | −2.8 | **1.1E-21** | **0.10** | −2.8 | **8.5E-14** | 0.87 | −1.6 | **2.1E-18** | 0.75 | −1.4 | **7.1E-16** |
| rs3732084 | ZDBF2 | 0.17 | −1.6 | **1.2E-28** | 0.14 | −2.3 | **1.9E-19** | **0.02** | −1.5 | **5.7E-17** | 0.53 | −1.6 | **2.0E-21** | 0.46 | −1.5 | **3.8E-19** |
| rs1975597 | ZDBF2 | 0.27 | −1.5 | **3.6E-29** | **0.08** | −2.2 | **4.0E-19** | 0.15 | −1.5 | **9.9E-17** | 0.52 | −1.4 | **5.4E-22** | 0.57 | −1.3 | **1.2E-19** |
| rs2585 | IGF2 | 0.17 | −1.1 | **5.9E-33** | **0.02** | −2.3 | **3.0E-22** | **0.08** | 0.2 | **1.3E-35** | 0.44 | −1.3 | **7.3E-15** | **0.10** | −2.3 | **6.1E-29** |
| rs7873 | IGF2 | 0.43 | −1.2 | **2.9E-33** | 0.14 | −2.4 | **3.9E-22** | **0.03** | 0.1 | **5.3E-36** | 0.78 | −1.4 | **2.1E-15** | 0.92 | −2.3 | **1.8E-28** |
| rs6059869 | HM13 | 0.17 | 0.3 | 8.3E-02 | **0.10** | 0.0 | 3.8E-01 | 0.13 | 0.2 | 4.0E-01 | 0.68 | 0.3 | **3.1E-02** | **0.03** | 0.6 | **7.2E-03** |
| rs6059873 | HM13 | **0.06** | 0.4 | **3.2E-03** | **0.02** | 0.2 | 2.8E-01 | **0.02** | 0.4 | 9.1E-01 | 0.33 | 0.4 | **1.5E-03** | **0.02** | 0.6 | **5.6E-04** |
| rs8110538 | ZNF331 | 0.27 | −0.9 | **3.2E-25** | 0.63 | −1.1 | **9.1E-18** | **0.08** | −1.0 | **4.5E-16** | 0.75 | −0.9 | **4.3E-19** | **0.08** | −0.8 | **3.3E-15** |
| rs8110350 | ZNF331 | 0.30 | −0.9 | **3.5E-25** | 0.44 | −1.0 | **5.8E-18** | **0.08** | −1.0 | **1.0E-15** | 0.81 | −0.9 | **1.4E-19** | **0.10** | −0.7 | **2.1E-14** |
| rs7810469 | PEG10 | 0.19 | 1.0 | **8.3E-03** | **0.04** | 0.9 | 1.2E-01 | 0.58 | 1.2 | 1.2E-01 | 0.53 | 1.0 | **2.8E-04** | **0.10** | 0.9 | **8.3E-03** |

[a]FDR-adjusted p-values are listed in column DI_p with significant results (FDR ≤ 0.1) shown in bold. Also log-fold changes (logFC) and FDR adjusted p-values of differential expression analysis (DE_p) are listed with significant results (FDR ≤ 0.05) shown in bold

of RNA- and DNA-based genotypes was successful (i.e. re-expressed allele agreed with DNA-based genotype), though complicated due to the low coverage WES data (Supplementary Fig. 14).

**Differential expression of imprinted genes**. In 2015, Kim et al. found 21 of their 23 (91%) analysed putatively imprinted genes to be differentially expressed in breast cancer[22]. As they had compiled imprinted genes from literature irrespective of tissue type, we performed differential expression (DE) analysis in control vs tumour and control vs breast cancer subtypes for the here detected imprinted SNPs and genes. Significant DE was found in almost all (92%) of the imprinted SNPs for all tumour samples. Imprinting is thus indeed heavily deregulated in breast cancer (Supplementary Tables 7 and 9). Far more loci were downregulated than upregulated: 87% were detected with a negative log-fold change. The FDR-adjusted p-values and log-fold changes of all SNPs showing DI can be found in Table 2 (Supplementary Data 5 and Supplementary Table 8 show results for all SNPs and genes, respectively).

One would expect that DI, i.e. relative higher expression of the silenced allele, implies upregulation of the imprinted gene as in canonical LOI. However, at least in breast cancer, DI was not associated with overexpression of the corresponding gene. The only clear exception was HM13, for which DI implied higher expression in most subtypes (and IGF2 in BL and PEG10 in LumB to a lesser extent, Table 2) in line with canonical LOI. For most other SNPs, DI was associated with expression downregulation.

We verified these results in heterozygous samples, as DI cannot be observed in homozygotes (Supplementary Note 5). Only for 5 SNPs (located in ZNF331, HM13, USP32P2, ZDBF2 and H19) sufficient WES data were available to accurately evaluate DI and DE results. In ZNF331, significant downregulation was found in biallelically expressing tumour samples compared to monoallelically expressing control samples, but also when compared to monoallelically expressing tumour samples. HM13, on the other hand, was significantly upregulated in biallelically expressing samples (compared to monoallelically expressing tumours as well as monoallelically expressing controls), further confirming the results presented above. For the other SNPs no differences were

detected, particularly due to low power (insufficient WES coverage).

**Residual biallelic expression as potential cause of DI**. Previous studies suggested that the basic concept of LOI in which re-expression of the imprinted allele leads to higher expression of imprinted genes is incomplete. Also here we found that DI is particularly associated with expression downregulation.

We hypothesise that DI/apparent LOI may also be caused by the presence of residual biallelic expression, i.e. incomplete silencing of the imprinted allele, cf. IGF2 Fig. 1a. If expression of the normally active allele is downregulated, expression levels for both alleles become more similar, which can be incorrectly perceived as LOI. To evaluate this hypothesis, the expression of the normally silenced allele (i.e. allele with least expression) was used as a measure for LOI. Results indicated no significant DE between cancer and controls of the silenced allele, supporting the hypothesis that the perceived LOI in breast cancer is particularly a by-product of the silencing of the active allele. Nevertheless, also the low number of biallelically expressing samples and thus decreased power may have an impact. It should be noted that for HM13—clearest example of an upregulated imprinted gene—a significant (unadjusted) p-value was observed for SNP rs6059873 (higher expression of the silenced allele) whereas this was not the case for the other imprinted loci (Supplementary Table 14).

Subsequently, we evaluated whether CNV aberrations may explain imprinted gene silencing (and thus DI) in breast cancer. Contrasting controls, significant associations were found between imprinted gene expression and CNV in cancer samples for 17 of the 23 genes present in TCGA CNV data (Supplementary Fig. 24, Supplementary Table 12 and Supplementary Data 6). For example, in cancer, GNAS, BCR and SNRPN showed both expression downregulation and CNV losses, while for HM13, gains were linked with overexpression. As we did not observe this for all loci and samples, we also evaluated DNA methylation. For most imprinted genes downregulated in cancer (15 of the 16 covered in TCGA), probes differentially methylated between cancer and controls were found. However, though these included many probes located in known DMRs and/or featured by putative monoallelic methylation in controls (particularly in MEG3 and MEST loci, over 25% methylation difference in DMRs), there was no clear enrichment for putative imprinting regulating loci

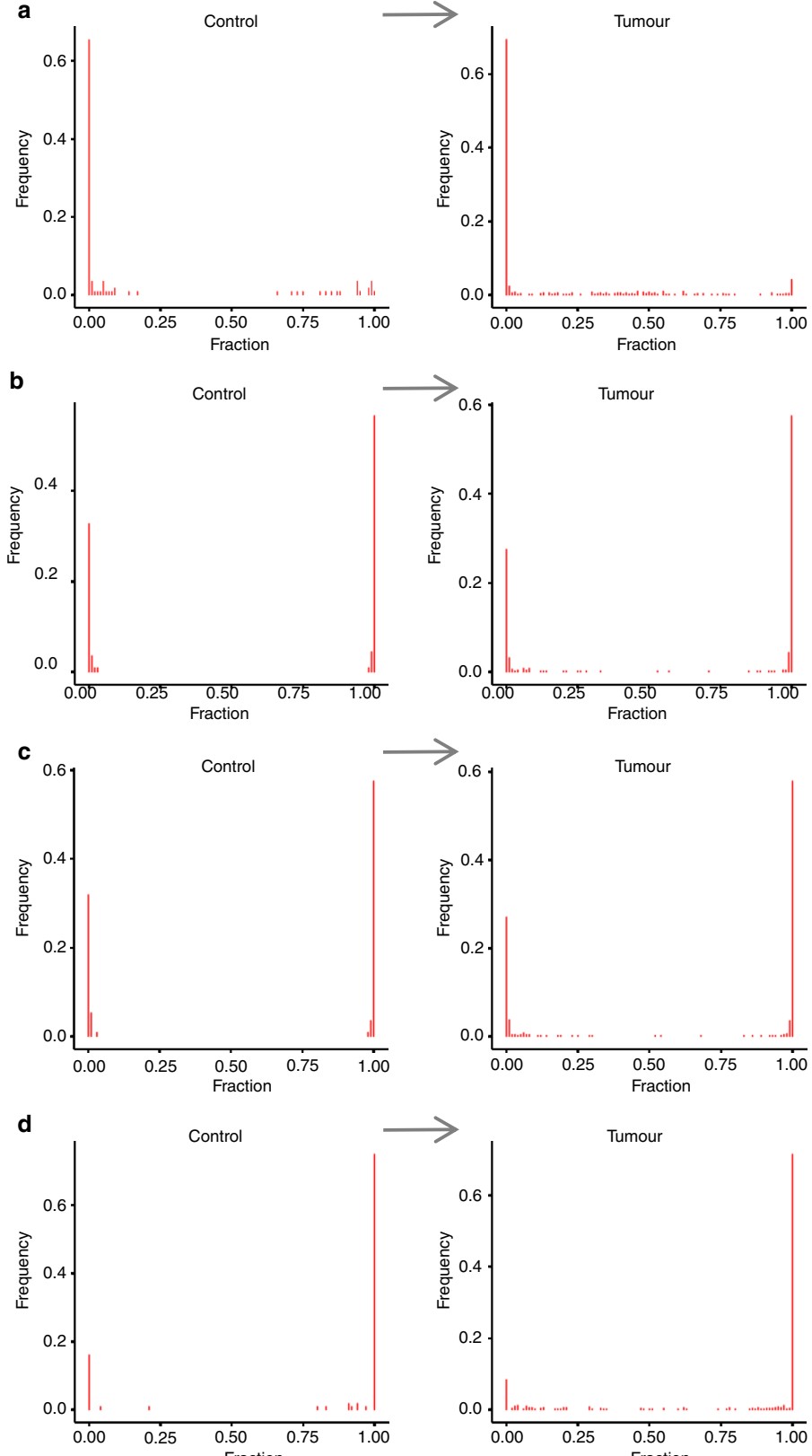

**Fig. 2** SNP positions differentially imprinted between normal and cancer samples. **a** *MEST* (rs10863, adj. *p*-value = 0.022). **b** *H19* (rs2839704, adj. *p*-value = 0.069). **c** *H19* (rs2839703, adj. *p*-value = 0.082). **d** *HM13* (rs6059873, adj. *p*-value = 0.062)

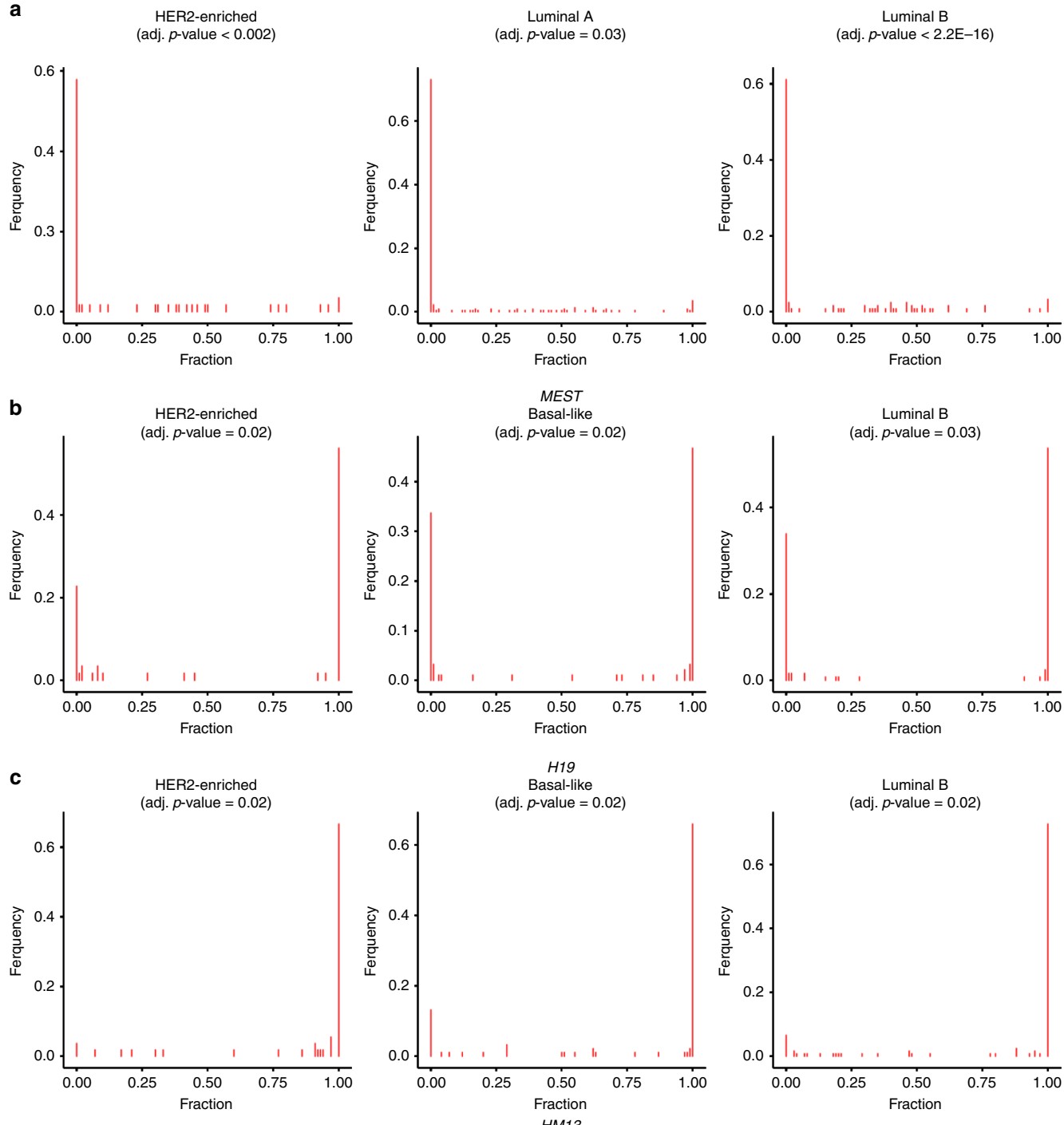

**Fig. 3** SNP positions differentially imprinted between normal and cancer subtypes. **a** *MEST* (rs10863). **b** *H19* (rs2839704). **c** *HM13* (rs6059873)

among differentially methylated probes (Supplementary Data 3). These results suggest that imprinted gene silencing in breast cancer is common and often associated with CNV and deregulated methylation, and is the most likely cause of DI in cancer (rather than its consequence).

**LOI, DE, CNV and differential methylation of HM13/MCTS2P.** *HM13* is the only gene in which both DI and higher expression in cancer was identified (Table 2), indicating LOI. However, not all SNPs in this gene showed consistent DI and DE results. 80 SNPs were initially analysed in the full *HM13* gene, yet

only 5 SNPs were maintained upon initial data filtering (see Methods, section Detection of imprinting and Supplementary Data 1). Of all SNPs detected to be imprinted, 4 located in exon 3 of *HM13* transcript 4, and a 5th one intronic in *HM13* but exonic in *MCTS2P* retrogene (Supplementary Fig. 27). Of the 4 detected imprinted SNPs in exon 3 of transcript 4, 3 demonstrated DE and 2 also DI (particularly in LumB), whereas the 5th SNP (exonic in *MCTS2P*) did exhibit DE but no DI. For SNPs featured by significant DI, we estimated that over 50% of all samples may be affected (Supplementary Table 5). Moreover, *HM13* DI was independent of lymphocyte infiltration (Supplementary Note 8). Subsequently, it was evaluated whether there was evidence for

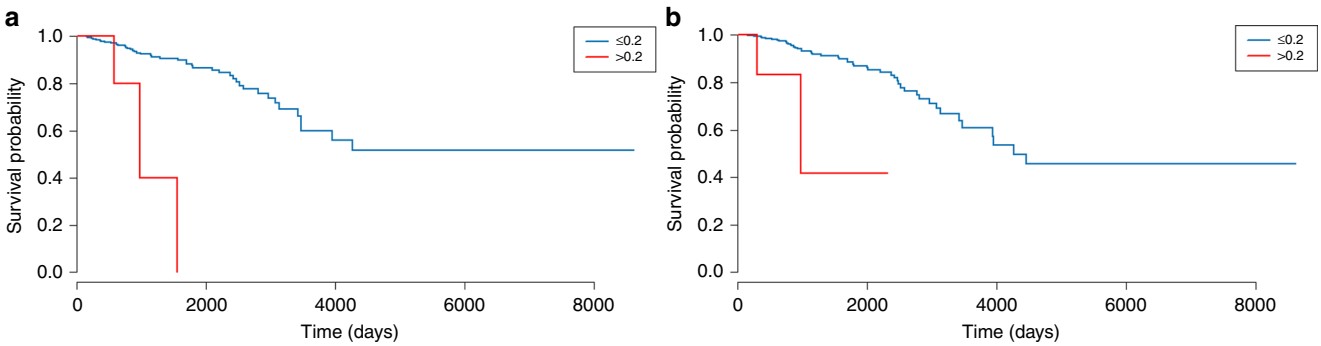

**Fig. 4** Kaplan–Meier plot of the Cox proportional hazards model for survival as a function of differential imprinting. Differential imprinting is implemented continuously as the allelic ratio (AR) of the least expressed allele over the most expressed allele (with AR as categorical variable for the Kaplan–Meier plot: AR ≤ 0.2 (blue curve) and AR > 0.2 (red curve)), and age. **a** rs3732084 (*ZDBF2*) **b** rs1975597 (*ZDBF2*)

transcript-specific DE, as the other exons lack the informative SNPs required to evaluate DI directly. All exons were significantly upregulated in breast cancer (subtypes), suggesting no transcript-specific effects (Supplementary Table 15).

Given that *HM13* is located on the 20q locus, often gained in cancer, we also evaluated whether CNV gains may explain DI, but DI appeared independent from CNV for all imprinted *HM13* SNPs (Supplementary Table 13). Subsequently, ten Infinium HumanMethylation450k probes demonstrating approximately 50% methylation (see Joshi et al. Table S2 and MEXPRESS[33,34], Supplementary Fig. 28) and located in the *HM13* DMR ([32], except for cg18471488) were analysed in TCGA. Methylation was significantly lower for two probes (cg18471488—located near *HM13* promoter region—and cg24607140—located near *MCTS2P* and already associated with imprinting control of the latter gene[33]) in tumour compared to control samples (Supplementary Table 16). Also in the subtypes, methylation of cg18471488 was significantly lower. HER2 and LumB did not show any other differential methylation, while in BL methylation levels were lower for almost all probes (Supplementary Table 16). Only in LumA significantly higher methylation was found, which concurs with results mentioned higher (i.e. DE but no DI was found in LumA). Methylation of probe cg18471488 was significantly correlated with expression of the last exon of transcript 4 (possibly the UTR of this transcript) but also the full *HM13* gene in the whole dataset (control and tumour data, Supplementary Table 17). In summary, these results show that LOI of the *HM13/MCTS2P* locus is linked to DNA methylation aberrations, but that a more precise description is hampered by the resolution of the data at hand.

## Discussion

Genomic imprinting is important for normal development and growth. The genome-wide evaluation of imprinting deregulation in cancer and diseases is currently hampered by a lack of appropriate data-analytical strategies. We hence developed a new methodology for the genome-wide detection of imprinting and its deregulation. After application on breast tissue, we were able to detect many imprinted genes and confirm major deregulation of imprinted genes in breast cancer and the varying subtypes. Imprinted genes exhibited clear differential expression, particularly downregulation, in tumour samples. Strikingly, in HER2 and BL tumours, downregulation was associated with massive induction of biallelic expression. A similar pattern has been described earlier for individual genes, e.g. for *IGF2* in oesophageal cancer[24]. However, most likely, this is merely the result of a higher relative expression of the (not completely silenced)

imprinted allele due to downregulation of the expressed allele. Given that the imprinted allele itself is not affected, we refrain from using the term LOI, and proposed the more generic term differential imprinting to indicate a relative shift in allelic expression independent of the underlying cause. The sole example where DI could be attributed to LOI was *HM13*, which exhibited overexpression in cancer, particularly in LumB tumours, due to re-expression of the normally silenced allele.

We analysed RNA-seq data of 113 breast tissue samples and 127 putatively imprinted SNPs in approximately 35 genes were identified and used for further analysis. For 2 SNPs no dbSNP annotation could be retrieved, though later manual curation suggested rs2269621 to be located in the known imprinted gene *L3MBTL1* (Supplementary Fig. 11b). Validation in GTEx data was possible for 121 SNPs corresponding to 30 genes. Although our methodology cannot assess whether the expression status of each allele is indeed determined by the parent of origin (which would require unavailable trio data), visual evaluation of the different mixture distribution plots as well as the clearly significant adjusted *p*-values strongly support at least monoallelic expression. Also, note that all genes demonstrating relevance in breast cancer had been associated with imprinting before. Moreover, the observation that newly identified putatively imprinted genes demonstrate similar differential expression patterns as known ones provides more evidence for their imprinting status. Finally, the identified loci largely featured known imprinting associated DMRs[32].

Compared to the study by Baran et al., which only used 27 breast samples, 13 genes were detected by both Baran's and our method[9]. One gene (*PPIEL*) was not evaluated by our methodology due to low coverage, whereas imprinted SNPs for the second gene (*SNURF*) were detected yet annotated as the overlapping gene (*SNRPN*). Of the 19 genes detected by our method only, at least 7 are known imprinted genes, also detected by Joshi et al.[33], or genes located in the neighbourhood of imprinted genes. It should be noted that accuracy of these results largely depends on the accuracy of the underlying annotation. Nevertheless, inconsistent results between SNPs in the same gene were evaluated, and appeared to be mainly caused by technical reasons, i.e. low coverage of intronic SNPs, low goodness-of-fit to the model or low allele frequency, as demonstrated for *HM13* in Supplementary Data 1. On the other hand, interestingly, for *ZNF300P1*, *LOC100294145*, *ZNF331* and *GNAS-AS1/GNAS*, we observed transcript-specific imprinting to be a more likely explanation.

Subsequently, we analysed 506 breast cancer samples for DI. One gene, *MEST*, showed significant DI and two genes, *H19* and *HM13*, were borderline significant. *MEST* is already known to

show DI in varying cancers, including breast cancer[21,35–37]. Aberrant *H19* imprinting is often seen in cancer as well and is thought to have an important role in cancer development[21,38,39]. When taking into account the different tumour subtypes, we detected 24 SNPs (in eight genes: *ZDBF2*, *PEG10*, *MEST*, *H19*, *IGF2*, *MEG3*, *ZNF331* and *HM13*) exhibiting DI in at least one subtype compared to the normal tissue samples. DI was particularly present in BL and HER2 tumours. Many of these genes had been linked with DI and cancer development before[21,35,40–43]. DI was typically not associated with survival, except for 2 SNPs in *ZDBF2*, a zinc finger-containing protein, yet this finding requires additional validation.

We found most of the imprinted genes to be downregulated in tumour samples, often leading to DI (but not canonical LOI). As recently demonstrated, particularly CNV was associated with expression deregulation of imprinted genes[27]. Methylation was also found to be significantly different between tumours and controls, though without clear enrichment for imprinting associated DMRs. For these loci, DI is thus most likely the consequence of transcriptional silencing. Previously, studies in murine and human prostate tissue and cancer detected a transcriptional network of co-regulated imprinted genes[26,44], with *PLAGL1* as putative key player[26,45–47]. Results from preliminary analyses (Supplementary Note 7) are compatible with the presence of an imprinted gene network in cancer as well, but an elaborated co-expression analysis is required to formally test this.

Contrasting other loci, *HM13* clearly exhibited canonical LOI, i.e. re-expression of the normally silenced allele leading to overexpression in cancer, particularly in the LumB subtype. Both significant SNPs appear to be present in the third (and last) exon of transcript variant 4, possibly the UTR of this transcript. Though other evidence for imprinting of *HM13* exists, only little information is available on its function—a signal peptide peptidase involved in the immune system[48,49]. Importantly, deregulation of *HM13*—located on 20q—has been revealed in colorectal carcinoma: the often observed 20q gain in this tumour is associated with higher *HM13* expression, which was demonstrated to lead to accelerated growth of the tumour[50]. Also in the study at hand, CNV gain led to increased *HM13* expression, yet this was independent of LOI. Interestingly, in an imprinting study in normal blood samples by our group, *HM13* also appeared to be featured by LOI and higher expression in a subset of samples, particularly in older individuals (unpublished results, https://bit.ly/2oCR6eD). With respect to the mechanism of *HM13* deregulation in breast cancer, we demonstrated that aberrant methylation in the neighbourhood of the *HM13* promoter was linked to deregulation of its expression. Differential methylation may lead to different polyadenylation (as described in mice[51]) and hence varying transcripts, yet we found differential expression of all exons in *HM13*. Our methodology also did not allow to exclude promoter switching from an imprinted promoter to a non-imprinted one, as already described for *IGF2* and *MEST*[27,36], as cause of *HM13* LOI in cancer. Further research is, therefore, necessary to unravel the exact mechanism(s) and consequences of (de)regulation of *HM13* in breast cancer.

Throughout the manuscript, results were verified by comparison with available WES-based genotyping data. However, low or absent coverage of WES for the corresponding loci led to a massive loss of imprinted SNPs that could be evaluated. This further underscores the benefits and more general applicability of the introduced methodology, which solely focuses on RNA-seq data. Also, this may explain why novel imprinted loci were found in our analysis, compared to methods where genotyping data is used to identify heterozygous samples prior to detecting imbalanced allelic expression in the latter, e.g. Baran et al.[9]. Some methodological improvements could however further increase sensitivity and specificity. For example, the current method relies on Hardy–Weinberg equilibrium, but could be modified to take into account population substructure. Additionally, the current mixture of binomial distributions could be updated to a mixture of beta-binomial distributions, as the latter captures more natural variation in expression between alleles. Nevertheless, the current study (cf. Figure 1 and Supplementary Figs. 1–11) clearly demonstrates that the proposed methodology is sufficiently robust. Another putative limitation of this study is the fact that tumour impurity, e.g. by infiltrating lymphocytes, may lead to the erroneous conclusion of LOI. This may be particularly relevant for *HM13*, given that *HM13* is expressed in blood and that admixture of biallelically expressed *HM13* could theoretically lead to both higher *HM13* expression and LOI in cancer. Nevertheless, we have previously found imprinting of *HM13* in blood (unpublished results, https://bit.ly/2oCR6eD), and could not find a significant correlation between infiltrating lymphocytes and *HM13* LOI, thereby rejecting confounding due to lymphocyte infiltration. As a final limitation, it is important to note that no distinction between primary and secondary imprints is possible with our methodology.

In conclusion, this study demonstrates that imprinting is indeed heavily deregulated in breast cancer, though the mechanism of its deregulation is complex. Many imprinted genes are downregulated in cancer, likely leading to DI without actual higher expression of the silenced allele. One clear exception was found, *HM13*, with LOI and upregulation in cancer samples. We were able to detect these putatively imprinted genes and their deregulation with a newly developed method solely based on RNA-seq data. The effectiveness of our novel methodology and the advantage of solely using RNA-seq data was hence also confirmed.

## Methods

**Data.** RNA-seq data of 113 human healthy control and 506 diseased samples (only those for which a PAM50 subtype was available) of the TCGA breast invasive carcinoma dataset were used in this study[52]. RNA-seq BAM-files were downloaded from the prior TCGA data portal (https://portal.gdc.cancer.gov/legacy-archive/search/f). Invasive ductal carcinoma, which starts in the milk ducts of the breast, and invasive lobular carcinoma, which originates in the lobules, were both studied[28]. For all cancer samples, additional expression subtypes based on the PAM50 classifier were obtained from the UCSC cancer genome browser (8 NL, 92 BL, 228 LumA, 121 LumB and 57 HER2)[52]. In all of these samples, variants were called using Samtools mpileup/bcftools (v0.1.19)[53]. Female breast samples (92 samples) from GTEx data (in dbGaP under accessions phs000424.v6.p1) were used for validation[29].

**Significance threshold.** Throughout the manuscript, the Benjamini–Hochberg procedure was used for false discovery rate estimation[54]. For detection of imprinting and differential expression analysis, an FDR of 5% was used as significance threshold. For analyses where loss of power is anticipated due to non-informative homozygous samples (e.g. differential imprinting detection), an FDR of 10% was used.

**Genotype calling.** Genotype probabilities and corresponding nucleotide-read/ sequencing error rates were calculated from RNA-seq data using SeqEM (v1.0)[55], a fast Bayesian genotype-calling algorithm based on the expectation maximisation (EM) algorithm to estimate the prior allele frequencies and the nucleotide-read error rate in an iterative way. Note that imprinting biases RNA-seq-based genotyping (i.e. less heterozygous samples will be detected), yet that allele frequency estimates are unbiased as both alleles have an equal chance to be imprinted. Genotyping with SeqEM was done without the HWE option, only for estimation of the number of biallelically expressing samples in RNA-seq data HWE was assumed (Supplementary Note 4b).

**Detection of imprinting.** Rationale: The rationale behind the proposed methodology is that biallelic expression yields RNA-seq data (or other similar sequencing data) that is in HWE for each locus, i.e. if SNPs are present for a locus, both homozygous and heterozygous subjects will be detected at a predictable rate (under HWE assumptions)[56]. However, in case of monoallelic expression, heterozygous samples will no longer be detected in RNA-seq data resulting in deviation from the HWE, which can be measured (Fig. 5a). So for imprinted loci, no distinction can be

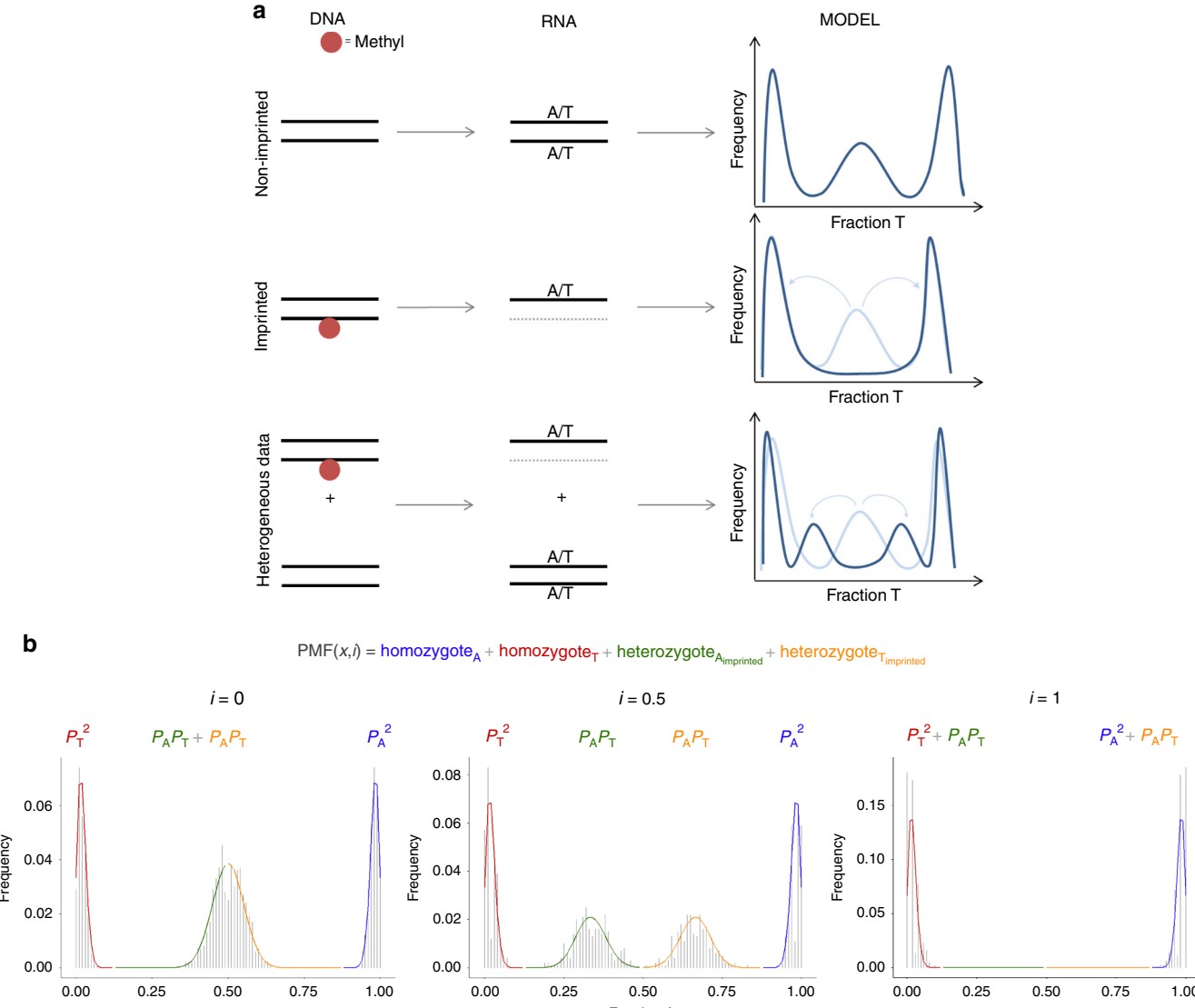

**Fig. 5** Graphical representation of rationale of the PMF. **a** The PMF is defined as a mixture model of genotype-dependent binomial distributions and describes the probability of observing specific RNA-seq coverages for each allele for a specific SNP locus. In these binomial probabilities, sequencing error rates, degree of imprinting (*i*) as well as the specific genotype are taken into account. For non-imprinted loci, the PMF results in two homozygous peaks and one heterozygous peak. For imprinting, on the other hand, no heterozygous can be detected on RNA-level and this peak is hence eliminated. Heterogeneous data leads to the detection of partial imprinting. **b** PMF for different degrees of imprinting. In this mixture model, the genotype-dependent binomial distributions have weights corresponding to their Hardy–Weinberg theorem derived expected chances

made between homozygous and heterozygous samples. Throughout the Methods section, we assume a locus with two alleles, A and T (with allele frequencies $P_A$ and $P_T$, respectively), yet this of course applies to all possible nucleotides.

Note that the same is true for enrichment-based sequencing data. Indeed, monoallelic histone modifications, as well as monoallelic DNA methylation, lead to ChIPseq and MethylCap-seq data, respectively, that is no longer in HWE, e.g.[30].

To enable screening for loci featuring imprinting, a probability mass function (PMF) describing the probability of observing specific coverages for each allele for a specific SNP locus was developed. As the probabilities depend on the underlying genotypes, the PMF was created as a mixture model of genotype-dependent binomial distributions with weights corresponding to the probabilities under Hardy–Weinberg (Fig. 5b)[56]. Sequencing error rates (median per chromosome) are here taken into account. Subsequently, maximum likelihood estimation was used to estimate the degree of imprinting (i) and a likelihood ratio test was constructed to detect significant imprinting. A detailed discussion of the different elements of this PMF and the analytical framework can be found in the following subsections. All analyses were performed in R (v3.3.2)[57], scripts are available upon reasonable request.

Data filtering using an empirical Bayes approach: After SNP calling, for each SNP position samples were filtered and corrected to obtain nucleotide sequences containing a maximum of two alleles. Using a maximum of two alleles per locus

ensures high-quality sequences, but was also a prerequisite for the genotype calling step later on (by SeqEM). As dbSNP was used for SNP calling, the dbSNP alleles were chosen as the two standard alleles. In case dbSNP contained three or more alleles for a particular SNP position, the standard alleles were chosen as the two dbSNP alleles with the highest mean allele frequencies for that locus over all samples. A quality filtering procedure was included to retain only those samples featuring one (homozygous) or both (heterozygous) standard alleles (= reference alleles defined by dbSNP for that particular SNP position). By default, samples already containing the two standard alleles as most frequent alleles were retained, whereas samples for which the allele with the highest frequency was not a standard allele were filtered out. However, for samples characterised by a non-standard allele as the second most abundant allele, an empirical Bayes approach was implemented to filter out putative heterozygous samples (one standard allele and one non-standard allele) yet keeping homozygous samples (one standard allele and sequencing errors). This procedure goes as follows:

i.  The posterior probability of obtaining a specific observation given a heterozygous sample was determined using a multinomial distribution:

$$P(\text{data}|\text{heterozygous}) = M(\text{data}; x_1, x_2, x_3, x_4) \qquad (1)$$

with both $x_1$ and $x_2$ equal to the sum of the two highest allele frequencies of the

sample divided by two, and $x_3$ and $x_4$ are equal to the sum of the two lowest sample allele frequencies divided by two. Note that the two alleles corresponding to these latter two frequencies could only be obtained due to sequencing errors.

- The probability that the sample is heterozygous is calculated as:

$$P(\text{heterozygous}) = 2pq \tag{2}$$

with $p$ and $q$ the mean allele frequencies over all samples of the standard and non-standard allele, respectively.

ii. Similar to the approach for heterozygous samples, the posterior probability of the obtained nucleotide sequence in case of a homozygous sample is again defined with a multinomial distribution:

$$P(\text{data}|\text{homozygous}) = M(\text{data}; x_1, x_2, x_3, x_4) \tag{3}$$

However, here only one allele is genuine while the other three are the result of sequencing errors. Therefore, $x_2$, $x_3$, $x_4$ are determined as the sum of the lowest three allele frequencies divided by three, while $x_1$ represents the allele frequency of the 'true', standard allele (having the highest sample allele frequency).

- The probability of the sample originally being homozygous is next defined as:

$$P(\text{homozygous}) = p^2 \tag{4}$$

as in Eq. (2) $p$ again represents the mean allele frequency of the standard allele over all samples.

iii. Finally, following criterion was used to identify putative heterozygous samples:

$$P(\text{heterozygote}|\text{data}) \geq P(\text{homozygote}|\text{data}) \tag{5}$$

Using Bayes' theorem and knowing that $P(\text{data})$ is equal for both $P(\text{heterozygote}|\text{data})$ and $P(\text{homozygote}|\text{data})$, this can also be written as:

$$P(\text{data}|\text{heterozygous}) * P(\text{heterozygous}) \geq P(\text{data}|\text{homozygous}) \\ *P(\text{homozygous}) \tag{6}$$

Next, identified putative heterozygous samples that include non-standard alleles are removed from the dataset for the locus under study.

Genotype calling and filtering: Calculation of genotype probabilities and corresponding nucleotide-read/sequencing error rate was done using SeqEM (v1.0), a fast Bayesian genotype-calling algorithm based on the expectation maximisation (EM) algorithm to estimate the prior genotype frequencies and the nucleotide-read error rate in an iterative way.

As model-based approaches (such as SeqEM) are prone to false positives due to pre-processing induced artefacts (e.g. alignment errors), only reliable loci were used for further analysis. The standard approach of filtering is based on HWE and not applicable here, leading to the necessity to use a combination of alternative criteria. Therefore, after obtaining estimates of both the allele frequencies and sequencing error rate, the SNP loci were subjected to extra filtering steps: (i) minor allele frequency > 0.1, (ii) median coverage > 4, (iii) estimated sequencing error rate ≤ 0.035 and finally, (iv) the number of samples covering the specific locus had to be at least 75. Loci which successfully passed these previous filters were subsequently tested with two final quality checks based on 'goodness-of-fit' criteria.

Additional data filtering using a goodness of fit procedure: SNPs that already passed the basic filtering steps were subjected to two final checks based on two methods assessing their goodness-of-fit in the model, independent of the presence of (partial) imprinting. This approach is particularly aimed to remove loci exhibiting good sequencing characteristics, yet with properties that indicate deviation from the standard genetic models. This may be due to technical (e.g. mismapping) but also biological (e.g. presence of SNP dependent expression differences) reasons.

As a first control, the $\chi^2$ test is used for goodness-of-fit. Though the exact distribution depends on the level of imprinting, this is not the case for the fractions of samples with respectively a higher reference allele (expected $P_A^2 + P_A P_T$) resp. variant allele count (expected $P_T^2 + P_A P_T$). Upon comparison of the observed and expected sample counts using the $\chi^2$ test, only loci with a resulting $p$-value > 0.05 were retained.

Next to a $\chi^2$ test, likelihoods are also a benchmark for goodness-of-fit. As likelihoods strongly depend on coverage (e.g. impact binomial coefficient), the likelihood of each individual measurement was multiplied by its coverage+1. The mean of these corrected log-likelihoods was next used as measure for the goodness-of-fit: loci with a mean ≤ 1.2 were filtered out. Though empirical in nature, this

filter setting largely removed remaining loci featured by aberrant allelic distributions.

After prior filtering of the data, the remaining SNPs were screened for imprinted regions by a likelihood ratio test (LRT).

PMF calculation: First, the probability mass function describing the probability of observing specific coverages for each allele for a specific SNP locus was established. As the probabilities depend on the underlying genotypes, we established the PMF as a mixture model of genotype-dependent binomial distributions with weights given by the expected probabilities under Hardy–Weinberg equilibrium (Fig. 5a). Importantly, in these binomial probabilities, sequencing error rates, degree of imprinting as well as the specific genotypes are taken into account. Ultimately this leads to the following PMF for e.g. a locus with two alleles A and T (equivalent for any other combination of two alleles) (Fig. 5b):

$$\text{PMF}(x, i) = P_A^2 B(x; p_A = 1 - \text{SE}, p_T = \text{SE}) + P_T^2 B(x; p_A = \text{SE}, p_T = 1 - \text{SE})$$
$$+ P_A P_T B\left(x; p_A = \frac{0.5 - i/2}{1 - i/2}(1 - \text{SE}) + \frac{0.5}{1 - i/2}\text{SE}, p_T = \frac{0.5}{1 - i/2}(1 - \text{SE}) + \frac{0.5 - i/2}{1 - i/2}\text{SE}\right)$$
$$+ P_A P_T B\left(x; p_A = \frac{0.5}{1 - i/2}(1 - \text{SE}) + \frac{0.5 - i/2}{1 - i/2}\text{SE}, p_T = \frac{0.5 - i/2}{1 - i/2}(1 - \text{SE}) + \frac{0.5}{1 - i/2}\text{SE}\right) \tag{7}$$

With

$x$ the coverages for alleles A and T, i.e. $x = (n_A, n_T)$
$P_A$ and $P_T$ the estimated population allele frequencies for a specific locus over all samples (obtained by SeqEM)
SE the estimated sequencing error rate (median per chromosome, obtained by SeqEM)
$i$ the degree of imprinting (varying from not ($i = 0$) to fully ($i = 1$) imprinted)
$B(x; p_A, p_T)$ the binomial probability for $x$ given the probabilities for each allele, i.e. $p_A$ and $p_T$, which depend on the specific genotype, SE and imprinting factor $i$

One already familiar with binomial distributions will notice that this is a somewhat alternative representation than typically used. However, it is simple to see that here, the chance of 'success' is represented by $p_A$ (indeed, $p_A + p_T = 1$) whereas the 'total number of trials' equals $n_A + n_T$.

Note that this PMF can be easily extended towards four alleles by considering a mixture of multinomial instead of binomial distributions. However, for simplicity - and as SeqEM can only handle two alleles per locus—we considered only two alleles.

From a practical point of view, the binomial coefficient is identical for each binomial distribution (within a sample, not between samples) and expressed as:

$$b = \frac{(n_A + n_T)!}{n_A! n_T!} \tag{8}$$

As ideally only a single allele is observed for homozygous samples, potential imprinting cannot be deduced from the allelic coverages. Here, the binomial probability will depend only on the SE - which is obtained by SeqEM. Because this error rate is assumed to be equal for all loci but can be ill-estimated when imprinting is present, the median SE over all loci is used. For homozygote AA, for example, the chance of observing allele A ($= p_A$) is 1-SE, while the chance of observing allele T ($= p_T$) is equal to SE, as T can only be present in the data due to a sequencing error.

$P(n_A, n_T)$ then becomes:

$$\begin{aligned} P(n_A, n_T) &= B(x; p_A = 1 - \text{SE}, p_T = \text{SE}) \\ &= b p_A^{n_A} p_T^{n_T} \\ &= b(1 - \text{SE})^{n_A} \text{SE}^{n_T} \end{aligned} \tag{9}$$

In the PMF this value of $P(n_A, n_T)$ is multiplied by the probability of the sample being homozygous AA. Assuming Hardy–Weinberg equilibrium (HWE), this equals the respective population allele frequency squared, i.e. $P_A^2$.

For heterozygotes, for example AT, potential imprinting has to be taken into account. This is done by including an imprinting factor $i$ that can vary between 0 (no imprinting) and 1 (fully imprinted) and is estimated using Maximum Likelihood Estimation (MLE, see below). Without imprinting, in theory both alleles A and T will be expressed to a similar extent so that $p_A$ and $p_T$ can be set to 0.5. However, when imprinting is present, the probability of observing the imprinted allele diminishes with a factor $i/2$. As the probabilities for both alleles need to sum to one, both probabilities are normalised by dividing them by a factor $1 - i/2$ ($= 0.5 + 0.5 - i/2$). Also sequencing error rates have to be taken into account, as a fraction SE of the normalised probability of one allele will be observed as the other allele and vice versa. Thus, when allele A is imprinted, the probability of observing

allele A ($= p_A$) equals:

$$\frac{0.5 - i/2}{1 - i/2}(1 - \text{SE}) + \frac{0.5}{1 - i/2}\text{SE} \tag{10}$$

While for the probability of $p_T$ this becomes:

$$\frac{0.5}{1 - i/2}(1 - \text{SE}) + \frac{0.5 - i/2}{1 - i/2}\text{SE} \tag{11}$$

Leading to the following formula:

$$P(n_A, n_T | \text{A imprinted}) = b\left(\frac{0.5 - i/2}{1 - i/2}(1 - \text{SE}) + \frac{0.5}{1 - i/2}\text{SE}\right)^{n_A}\left(\frac{0.5}{1 - i/2}(1 - \text{SE}) + \frac{0.5 - i/2}{1 - i/2}\text{SE}\right)^{n_T} \tag{12}$$

However, the possibility of imprinting of allele T also has to be taken into account:

$$P(n_A, n_T | \text{T imprinted}) = b\left(\frac{0.5}{1 - i/2}(1 - \text{SE}) + \frac{0.5 - i/2}{1 - i/2}\text{SE}\right)^{n_A}\left(\frac{0.5 - i/2}{1 - i/2}(1 - \text{SE}) + \frac{0.5}{1 - i/2}\text{SE}\right)^{n_T} \tag{13}$$

Likewise as for homozygotes, the binomial probability for the heterozygous fraction has to be multiplied by the genotype frequency $2P_AP_T$. As from the underlying biology both alleles can be assumed to have an equal chance of imprinting ($= 50\%$), this ultimately leads to the mixture PMF denoted in Eq. (7).

Imprinting factor: In a next step, the degree of imprinting (or imprinting factor) $i$ for a specific SNP locus is estimated using MLE. The likelihoods are calculated as the sum of the logarithmic values of the PMF-derived probabilities (Eq. (7)) over all samples. In summary, for each locus, i is varied from 0 to 1 (step size $= 0.01$), retaining the value of $i$ corresponding to the highest likelihood $\left(\hat{i} = \text{ArgMax}_i \prod_{a=1}^{n} \text{PMF}(x_{a,i}) = \text{ArgMax}_i \sum_{a=1}^{n} \log\left(\text{PMF}\left(x_{a,i}\right)\right)\right)$. Hence, for every SNP locus a degree of imprinting is obtained.

Likelihood ratio test: Finally, in order to screen for imprinted loci a likelihood ratio test is performed. The respective null and alternative hypotheses for a locus are:

$H_0$: the locus is not imprinted
$H_1$: the locus is imprinted

With the previous definitions this translates into:

$H_0$: $i = 0$
$H_0$: $i > 0$

Thus, the null hypothesis of no imprinting ($i = 0$) is compared to the alternative hypothesis that the locus is imprinted ($i > 0$). Practically, in a first step, the PMF (Eq. (7)) for the locus under study is calculated with i equal to 0. Next, the PMF is determined with the estimated value of $i$ as explained in the previous section. The obtained PMFs are then used in a LRT:

$$\Lambda(X) = \frac{f(X|H_0)}{f(X|H_1)} = \frac{\mathcal{L}(H_0|X)}{\mathcal{L}(H_1|X)} = \frac{\text{PMF}(x_1, i=0)\text{PMF}(x_2, i=0)\dots\text{PMF}(x_n, i=0)}{\text{PMF}(x_1, i=i)\text{PMF}(x_2, i=i)\dots\text{PMF}(x_n, i=i)}$$
$$= \frac{\prod_{a=1}^{n}\text{PMF}(x_a, i=0)}{\prod_{a=1}^{n}\text{PMF}(x_a, i=i)} \tag{14}$$

As the null hypothesis is a special case of the alternative hypothesis, the test statistic for nested models $-2\ln(\Lambda)$ can be used. This test statistic is $\chi^2$ distributed and $H_0$ will be rejected if its value is greater than $\chi^2_\alpha$. However, because we are testing at the border of a constrained parameter space (i equal to 0), a mixture of $\chi^2$ distributions is used: under the null hypothesis, the test statistic is distributed as an equal mixture of two $\chi^2$ distributions, namely $\chi^2_0$ and $\chi^2_1$ with 0 and 1 degrees of freedom, respectively[58]. Finally, a locus is called imprinted if the corresponding FDR-corrected $p$-value was smaller than the nominal FDR level of 0.05 (Benjamini–Hochberg correction for multiple testing).

Median imprinting: A robust measure for degree of imprinting, called median imprinting, was developed to enable the identification of a robust set of imprinted loci from the significant set identified higher. For each SNP locus, sample-specific ratios ($= R_{i,s}$ for SNP i and sample s) are calculated as the lowest allele count over the highest allele count, yielding values between 0 (only one allele expressed) and 1 (both alleles expressed to an equal extent). Next, the values of these ratios are sorted over all samples and the value of the 'median putatively heterozygous sample (sm)' is calculated. This sample corresponds with rank round(samplesize*$(1-P_A-P_T-P_AP_T)$) = round(samplesize*$(P_AP_T)$). The median imprinting value is then calculated as 2* $(0.5-R_{i,\text{sm}})$. In a last step, SNP positions with a median imprinting level $\geq 0.8$ were considered as robust (TCGA). For GTEx (validation), we used a median imprinting level of 0.4 as cut-off, given that typical artefacts had already been eliminated.

**Detection of differential imprinting.** Next to the detection of imprinted loci, we also examined possible deregulation of imprinting in cancer. This was done by testing for different relative expression of both alleles, here termed DI. When associated with absolute re-expression of the originally silenced allele, this is coined LOI. Briefly, ratios of the lowest allele count over the highest allele count (i.e. $R_{i,s}$) are calculated for each single SNP, over all samples. These ratios are sorted per SNP in an ascending order separately for control and tumour samples. As the lowest ratios are expected for homozygous samples (ratios theoretically equal to 0, yet slightly higher due to the presence of sequencing errors), one can consider samples with the highest $2P_AP_T$ ratios as putative heterozygous samples for that specific locus. In practice: samples with rank higher than sample_size*$(P_A^2+P_T^2)$) are considered as the heterozygous samples for a specific locus. After determining the mean ratio of these heterozygotes ($R_{i,\text{tumour}}$ and $R_{i,\text{control}}$), parameter $R_{i,\text{diff}}$ is calculated as the difference between these values, i.e. $R_{i,\text{diff}} = R_{i,\text{stumour}} - R_{i,\text{scontrol}}$. Upon random assignment of the tumour and control labels to the present samples by permutation, 10,000 random values of $R_{i,\text{diff}}$ are simulated to generate a null distribution. Loci with an FDR-adjusted $p$-value smaller than the nominal 10% FDR level were concluded to be differentially imprinted between control and tumour samples. We were solely interested in DI in cancer and hence we exclusively tested for higher ratios in tumour compared to control data. The different breast cancer subtypes were also tested for DI (though the Normal-like subtype was not studied here, as only 8 samples were available) in which the $p$-value was corrected over all samples. Here, it should be noted that considering the ratios allows for detecting differences independent of alterations in expression levels of imprinted genes, which are also prominent in breast cancer[59].

Subsequently, for loci featured by DI, the latter was linked to survival. A continuous DI variable, which was defined as the allelic ratio (allele count least expressed allele/allele count most expressed allele), was associated with survival and adjusted for age with a Cox proportional hazard model. To anticipate assumption violations, a null distribution was constructed by 10,000 permutations by randomly shuffling the ratios over the samples. Loci with an FDR-adjusted $p$-value $\leq 0.1$ were called significant. The analysis was also performed on solely the putative heterozygous samples, meaning that the $2P_AP_T$ fraction of samples with the highest allelic ratios were used (see higher).

Differential expression of the silenced (lowest expressed) allele was analysed as well to assess whether expression of the normally silenced allele was higher in tumour data. Here, counts per million (CPM) reads of the least expressed allele count were calculated as described in the next paragraph. The same permutation test as mentioned higher yet based on the logCPM-values rather than ratios, was performed on the SNPs showing significant DI.

**Detection of differential expression.** DE analysis was performed to further evaluate deregulation of imprinted genes in cancer. EdgeR normalisation factors were calculated from the breast cancer RNA-seq expression count file downloaded from firebrowse.org[60]. Afterwards, CPM-values for the imprinted SNPs were computed with these normalisation factors and library sizes (available for 100 control samples and 469 tumour samples: 87 BL, 54 HER2, 210 LumA, 111 LumB and 7 NL). EdgeR-based DE analysis was developed to increase power, at the cost of several assumptions. As the sample size and thus power is sufficiently high in the case at hand, we opted to use more robust standard non-parametric methods. Differential expression in control versus tumour samples was analysed with a Wilcoxon Rank Sum test for detected SNPs as well as the corresponding genes (sum of the CPM-values of the matching SNPs were used). To test for DE in the different breast cancer subtypes, a Kruskal-Wallis test and Dunn's post-hoc test were performed on the CPM-values of the varying subtypes.

To study transcript-specific effects in *HM13*, logCPM-values of exonic expression data (of 14 exons, downloaded from firebrowse.org[60]) were normalised and analysed for DE between control and tumour samples as described in the previous paragraph. Exon data were available for 468 tumour samples (85 BL, 54 HER2, 211 LumA, 111 LumA and 7 NL) and 100 control samples. Exonic RPKM values, available from firebrowse.org as well, were used for additional verification and consistently yielded the same conclusions.

**CNV and methylation analyses.** Infinium HumanMethylation450k data were downloaded from firebrowse.org[60] and CNV data from the GDC portal of TCGA. Infinium methylation data (450k) could be retrieved for 84 control and 207 tumour samples (35 BL, 14 HER2, 107 LumA, 46 LumB and 5 NL), whereas CNV data was available for 92 control and 506 tumour samples. A Wilcoxon Rank Sum test was performed to screen for significant differential methylation in probes located in imprinted genes. In the CNV data, gains and losses were called as segment mean > 0.2 and < −0.2, cf.[61]. A linear model was constructed to model expression (gene-based logCPM counts) as a function of CNV (factor), adjusting for breast cancer subtype.

For *HM13*, also the link between CNV and LOI was of interest. A $\chi^2$ test comparing LOI/not LOI (based on genotype calling with SeqEM on RNA-seq data, see Supplementary Note 4c and Supplementary Table 5) with CNV was performed for all imprinted *HM13* SNPs. Subsequently, methylation of 10 probes (listed in Joshi et al., with additionally cg18471488 as identified using MEXPRESS in the breast cancer population[33,34]) and exonic expression data (see Methods, section Detection of differential expression) were correlated to identify which methylated

locus might control expression. Both expression and methylation data were available for only 72 control and 192 tumour samples. A Spearman correlation test was performed for detecting correlation between the β-values of the 10 probes and (i) logCPM-values of the whole HM13 gene (gene counts obtained from firebrowse. org[60] and normalised with EdgeR) and (ii) logCPM-values of the 3rd exon of transcript 4 (as most of the imprinted SNPs were located in the neighbourhood of this exon). Again, RPKM values were successfully used for verification.

**Quality control.** To additionally verify the imprinting and deregulation results, corresponding WES data (BAM files) were downloaded from TCGA to obtain the underlying genotypes. However, only for 93 control samples and 464 tumour samples WES data were available. Concordance between WES and RNA genotypes was examined to validate the quality of genotyping with RNA-seq data (Supplementary Note 3c).

**Code availability.** The scripts are available from the authors upon reasonable request.

## Data availability

The authors declare that all data supporting the findings of this study are available within the Article and its Supplementary Information or from the corresponding author on reasonable request. Access to controlled GTEx (phs000424) and TCGA (phs000178) data was obtained through the database of Genotypes and Phenotypes (dbGaP). The TCGA data were accessed through the Genomic Data Commons portal (https://portal.gdc.cancer.gov/ and https://portal.gdc.cancer.gov/legacy-archive/search/f) and Broad Institute's Firebrowse data portal (http://firebrowse.org/). GTEx data were obtained from the dbGaP database directly (https://dbgap.ncbi.nlm.nih.gov/).

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

## Acknowledgements

The authors would like to thank Gaetan De Waele, Daria Morozova, Ine Haudenhuyse and Guillaume Henderson for their assistance with the imprinted gene networks. The results shown in this manuscript are based upon data generated by the TCGA Research Network (https://cancergenome.nih.gov/) and the GTEx Project consortium (www.gtexportal.org). We are very grateful to these consortia and their funding agencies for creating these extremely valuable datasets and making them available for additional research.

## Author contributions

T.G. and S.S. contributed equally to this work. The statistical approach was conceived by T.D.M. and developed by T.D.M., T.G. and S.S. with additional biological and technical insight from W.V.C and O.T.; general strategy: T.G., S.S. and T.D.M., data pre-processing & management: T.G., S.S. and J.G. Implementation and data analyses were done by T.G. and C.A.V. under supervision of S.S. and T.D.M. Additional validation was done by S.S. and C.A.V. with contributions from T.G., T.D.M., T.G. and S.S. wrote the manuscript with input from T.D.M., J.G., C.A.V., W.V.C. and O.T. All authors read and approved the final text.

## Additional information

**Competing interests:** The authors declare no competing interests.

