## [Peer Review File · Nature Communications]

Reviewers' comments:

Reviewer #1 (Remarks to the Author):

Loss of imprinting is a common observation in cancer and it would be valuable to have a reliable method for looking at genomic imprinting as a class in relation to cancer and many other fields. This paper reports a new approach to this and it has identified some interesting and novel targets. The approach is generally well described and it seems logical but more could be done to check the findings. The paper doesn't go much beyond reporting the findings and describing the method. The authors refer to transcriptional networks when interpreting the data but don't go much deeper. It would have been reassuring to see some further work looking at transcriptional networks (e.g. by applying the C techniques or possibly accessing existing databases) to check the biological plausibility of the imprinted regions they have identified using this new technique and the network interpretation.

It is important to note that the 50% methylation signature characteristic of imprinting (100% and 0%) can be maintained whilst the expression by allele may not follow exactly this pattern (e.g. the regions controlling transcription may not be imprinted); the manuscript states that "in breast cancer, LOI is not associated with overexpression of the corresponding gene". This means that an approach solely based on RNAseq data may be informative of some aspects of imprinting but not all and most likely it will be necessary to also do methylation analysis to complete the picture. The authors compare some of their data with 450k array data but the array has quite sparse coverage of imprinted regions. In a method development paper it would have been reassuring if the authors had carried out their own methylation analysis in the regions they report on to allow them to interpret the results and their meaning.

It would have been more convincing to see the results interpreted in a way that highlighted the ability of the method to produce biologically interesting results. For example, there seems to be no distinction between primary and secondary imprints in the paper. Presumably the method is not able to distinguish between these but it is important to take the difference into account when interpreting the data.

The paper is well written and generally easy to understand but there are some important exceptions and some minor typos and errors in syntax and grammar: e.g. "We hence evaluated the association between LOI and DE into more detail."

The paper states that: "SNPs in the same imprinted gene did not always show consistent (in)significant results. This can typically be attributed to technical/power associated causes. For example, the SNPs displaying LOI frequently had a higher coverage than non- LOI SNPs in the same gene." This is an important issue that needs to be better explained; if I understand it correctly this could be an important limitation to the method.

Reviewer #2 (Remarks to the Author):

In this manuscript Goovaerts and colleagues describe the characterization of imprinting using RNA-seq data for breast cancers from the TCGA consortium. This is an extremely timely as there is currently much interest in genomic imprinting in cancer. Furthermore, they describe an informatics method, based on deviation of Hardy-Weinberg equilibrium, in that heterozygosity/biallelic SNPs will not be detected following RNA-seq, which could be applicable to other datasets without accompanying genotypes. The manuscript is well-written, easy to read and follow. It is certainly an interesting study, which after addressing all comments below will be worthy of publication and I am sure will be of interest to the readers of Nature Communications.

- The introduction is easy to read and informative for the subject. The authors correctly discuss the issues raised by recent LOI studies, i.e. LOI associated with down-regulation/no change in expression despite the prediction that there should be a doubling, and the down-regulation despite normal methylation. To emphasize that expression levels and LOI could be important in cancer biology, and that relying solely on methylation profiling maybe misleading (as a biomarker), the authors should reference the recent Nature Communication publication (doi: 10.1038/s41467-017-00639-9).
- For example, recent studies identified.... 7. This reference is not recent since it was published in 2005! Please rephrase the sentence or use a more recent publication.
- The authors describe a method for profiling monoallelic expression of imprinted genes and their deregulation in cancer without the need for genotype data. The rationale is that a perfectly imprinted gene will be monoallelically expressed and no heterozygous cases will be observed in RNA-seq, this resembling a homozygous sample. But how do you discriminate homozygous loci from monoallelically expressed genes, or is in fact the data a mixture of the two, with imprinting inferred due to the lack of heterozygous/biallelic genes? Please state this clearly.
- Of the 127 SNPs identified a possibly imprinted, mapping to 35 genes, the authors mention that 16 are already known as imprinted. However the list in Table 1 is inaccurate since SNHG14 and PWAR6 are known to be paternally expressed (also known as the UBE3A-ATS non-coding transcript from which IPW, PWARs and snoRNA and miRNAs are processed)(DOI: 10.1093/hmg/dd5130). Therefore 27 SNPs should be reassigned as "known imprinted" and the numbers altered throughout the text.
- Of the list of putative imprinted genes, at least three of them are pseudogenes (MTCO1P12, CPHL1P and MTRNR2L1). How can the authors be sure that they are detecting the pseudo-copy, as stated, or the original gene when using short-read RNA-seq??
- The authors describe the difficulty in assigning monoallelic SNPs to overlapping transcripts and mention the MCTS2P/HM13 locus. This is true since the MCTSP2 retrogene can splice into the last eight exons of HM13. Please add a reference for this statement (DOI: 10.1093/nar/gkq1230 or alike).
- The imprinting (and evolutionary lack of) is very complicated and the HM13/MCTS2P locus is the basis of the imprinting by alternative polyadenylation. The description of this locus (and ideally a Supplementary Figure showing isoforms, SNPs and imprinted/LOI transcripts) should be given (DOI: 10.1101/gad.473408 and doi: 10.1093/nar/gkq1230).
- Since many reading this work may not be familiar with the bioinformatics approach, an example of a biallelically-expressed gene should be included in Figure 1. When the authors discuss partial imprinting of IGF2 in reference to Figure 1, could they give %-estimated range (i.e. 95% from one allele)?
- How can "intronic" SNPs be used to assess allelic expression? Are they detecting the primarily, unspliced transcripts?
- When comparing the putative monoallelic genes with TCGA WES genotypes, please indicate the number that were actually heterozygous at the DNA level and monoallelically expressed.
- Please add a Y-axis label to all graphs.
- When discussing LOI, the authors need to be aware that MEST and HM13 have overlapping non-imprinted transcripts that may be highly expressed in the cancer sample and therefore "mimic LOI" whereas it is actually expression of a non-imprinted isoform. This promoter switching from imprinted to biallelic has been reported for MEST (PMID: 12023987 and discussed in doi: 10.1038/s41467-017-00639-9). The authors should ascertain if the samples with LOI MEST have high expression of the MEST variant_2 (NM_177524).
- The non-imprinted isoform of MEST is tissue-specific and expressed in blood only (DOI: 10.1086/302712). Do the authors see any correlation between infiltrating lymphocytes and the presence of MEST biallelic expression?
- The authors need to clarify whether the inconsistency described for SNPs within the same gene is actually due to technical/power associated causes as stated, or due to re-expression of overlapping non-imprinted isoforms as could occur for a large number of imprinted genes (PLAGL1, GRB10, MEST, MCTS2/HM13, INPP5F/INPP5Fv2, NAPIL5/HERC3, PPIEL, ERLIN2, ZNF331, NAA60, WRB and SNU13).

- For the LOI observed in HER2 and BL subtypes the number of individual samples showing biallelic expression for each of the 8 genes should be given.
- For the LOI at the H19/IGF2 locus, was biallelic H19 expression associated with higher levels of IGF2?
- The section describing down-regulation, rather than over-expression, is interesting and important. However, the section would be more informative if the number of under/over-expressing tumour samples per subtype were given in a supplementary table (or the data presented in a figure), the range of this aberrant expression and if it was associated with genomic deletions/amplifications.
- LOI is a very well-defined term. As mentioned in the introduction it is the re-expression of the normally silent allele. Therefore is it possible to introduce a new term for the interesting observation that biallelic expression was associated with down-regulation of the normally expressed allele so that both alleles are more equally expressed at a residual level and perceived as LOI?
- Why do the authors use different FDR cut-offs in Table 2 for LOI (≤ 0.1) and DE (≤ 0.05)? Surely it would be best to use 0.05 throughout, which would then highlight that in a heterogeneous group of breast tumours LOI may not be significant, but with the correct classification significant LOI groups are observed.
- When comparing the HM450k methylation data for MCTS2P DMR within the HM13 gene, the authors only describe correlations for individual probes and increased expression. Is this because the (average) methylation over the entire DMR did not show any association? The MCTS2P DMR contains 10 probes (doi: 10.1080/15592294.2016.1264561 and doi: 10.1038/s41467-017-00639-9) and a supplementary figure should be included showing the methylation for all MCTS2P DMR probes not just those with significant correlations.
- The authors describe several genes for which imprinting has not been confirmed (MTCO1P12, LINC01139, PAX8-AS1, PTX3, CPHL1P, ATP8A1, ZNF300P1, LOC100294145, HOTAIRM1, RP11-109L13.1, GLIPR1/KRR1, PLIN1, TPSB2, USP32P2, MTRNR2L1, BCR). Rather than adding to the list of conflicting genes, which often hamper and confuse subsequent studies, the authors should present data confirming allelic expression (not only in breast but any tissue) or germline methylation.
- No evidence for an imprinted breast-cancer IGN has been presented. Minimal evidence should be given, for example do deregulated samples show LOI for multiple genes? Otherwise please refrain from mentioning it in the abstract discussion.

Reviewer #1 (Remarks to the Author):

Loss of imprinting is a common observation in cancer and it would be valuable to have a reliable method for looking at genomic imprinting as a class in relation to cancer and many other fields. This paper reports a new approach to this and it has identified some interesting and novel targets. The approach is generally well described and it seems logical but more could be done to check the findings. The paper doesn't go much beyond reporting the findings and describing the method.

We would like to thank the reviewer for his valuable feedback. Many additional analyses were performed for this revised version, also to check the findings. For example, at the SNP level, we could validate ~97% of imprinted SNPs in GTEx data. Moreover, we have focused on transcript-specific imprinting and identified several likely candidates (Supplementary Results, Section 3d). Finally, we have added DNA methylation, copy number variation and imprinted gene network analyses to evaluate possible mechanisms explaining imprinting deregulation and LOI in breast cancer (Results Section and Supplementary Results, Section 6 and 7).

The authors refer to transcriptional networks when interpreting the data but don't go much deeper. It would have been reassuring to see some further work looking at transcriptional networks (e.g. by applying the C techniques or possibly accessing existing databases) to check the biological plausibility of the imprinted regions they have identified using this new technique and the network interpretation.

The main novelty of this manuscript is the fact that we have developed an innovative methodology that demonstrated the relevance of loss of imprinting in breast cancer, underscoring the added value to the field. Based on the high validation rate and the fact that only RNA-seq data is required, we are convinced that application of this method will allow researchers to elucidate the mechanisms of imprinting (de)regulation, a topic that remains largely unexplored in cancer. In the revised version, as proof of concept, we have explored existing TCGA data (CNV, DNA methylation, expression data for co-expression/gene networks) to screen for such mechanisms, demonstrating CNV aberrations to play an important role, whereas deregulation of methylation and an imprinted gene network are likely to be less important (Results Section and Supplementary Results, Section 6 and 7). Combined with the complex experimental validation required of the latter (e.g. by 4C or creation of knockdowns/outs, taking into account allele-specificity), we believe further work on the networks to be outside of the scope of the manuscript at hand.

It is important to note that the 50% methylation signature characteristic of imprinting (100% and 0%) can be maintained whilst the expression by allele may not follow exactly this pattern (e.g. the regions controlling transcription may not be imprinted); the manuscript states that "in breast cancer, LOI is not associated with overexpression of the corresponding gene". This means that an approach solely based on RNAseq data may be informative of some aspects of imprinting but not all and most likely it will be necessary to also do methylation analysis to complete the picture. The authors compare some of their data with 450k array data but the array has quite sparse coverage of imprinted regions. In a method development paper it would have been reassuring if the authors had carried out their own methylation analysis in the regions they report on to allow them to interpret the results and their meaning.

We only focused on expression data to be unbiased towards regulatory mechanisms of imprinting. Imprinting by definition refers to the transcription level, and (as stated by the second reviewer as well) there are other mechanisms than DNA methylation that are also relevant for imprinting regulation (such as histone modifications). However, analysis of the TCGA 450k data revealed that for all genes present on the array at least one hemimethylated probe was present (methylation between 0.33 and 0.66, Supplementary Table 7), and that far more hemimethylated probes were present for imprinted loci than for non-imprinted ones ($p=2E-175$). More methylation analyses would definitely be interesting, but given other possible mechanisms of regulation, they would not give more conclusive results.

It would have been more convincing to see the results interpreted in a way that highlighted the ability of the method to produce biologically interesting results. For example, there seems to be no distinction between primary and secondary imprints in the paper. Presumably the method is not able to distinguish between these but it is important to take the difference into account when interpreting the data.

As stated higher, throughout the manuscript, we have particularly highlighted the biologically relevant results with respect to breast cancer, and have (in the revised version) also explored possible mechanisms of deregulation. Furthermore, additional validation was performed to ensure high-quality of reported imprinted loci, and we have identified loci featured by transcript-specific imprinting, leading to more biologically relevant results in “normal” tissue as well.

We agree that our methodology does not allow to distinguish secondary from primary imprinting events, and have added this to the limitations section of the manuscript (in the Discussion). Yet, to the best of our knowledge, such a distinction can only be accurately made by comparing the expression and epigenetics of gametes with those from embryos (or later developmental phases), and/or by considering ZFP57 binding patterns, rather than by using an alternative methodology.

The paper is well written and generally easy to understand but there are some important exceptions and some minor typos and errors in syntax and grammar: e.g. “We hence evaluated the association between LOI and DE into more detail.”

We thank the reviewer for his remark. We have corrected this error, and performed additional proofreading.

The paper states that: “SNPs in the same imprinted gene did not always show consistent (in)significant results. This can typically be attributed to technical/power associated causes. For example, the SNPs displaying LOI frequently had a higher coverage than non- LOI SNPs in the same gene.” This is an important issue that needs to be better explained; if I understand it correctly this could be an important limitation to the method.

There are several important aspects that should be mentioned here. First, the methodology is particularly limited by the quality (and quantity) of the RNA-seq data. Though the sensitivity is far larger than previous approaches (which also required high-quality exome-seq data), low coverage RNA-seq data for certain loci (even within genes) impairs any methodology (garbage in = garbage out). Indeed, for SNPs covered by on average few reads (e.g. in introns), it is very hard to accurately discriminate mono- from biallelic expression (imprinting detection) and even harder

to quantify and statistically test discrepancies regarding mono- and biallelic expression between sample groups. Therefore, this limitation cannot be considered a drawback of the method as such.

On the other hand, our focus on the SNP level (rather than transcript level) is an actual limitation of the method, as a simultaneous analysis of all SNPs in a transcript could in theory enable discriminating coverage/technical problems from the presence of transcript-dependent imprinting for a gene. Yet, also here, this limitation is particularly imposed by the properties of current second-generation sequencing RNA-seq data (particularly short read lengths), where isoform identification and quantification remains a clear challenge as such without even taken into account SNPs & linkage. We consider our methodology therefore rather tailored to the input data, rather than featured by clear limitations. Nevertheless, in the future (cf. the advent of single-molecule sequencing data), we should indeed consider moving to the transcript level, this is now mentioned in the Discussion section.

Finally, we have extensively explored SNP differences within single genes (particularly for the detection of imprinted loci, Supplementary Results, Section 3d), leading to the identification of several loci likely featured by transcript-specific imprinting, but also to the degradation of a few loci to “candidate imprinted” (i.e. when we could validate imprinting in GTEx, but no rational explanation was possible for the inconsistencies between SNPs). For the detection of LOI in cancer, results were consistent if coverage was sufficiently high, yet transcript specific effects may be present. Indeed, for ZNF331, 2 LOI SNPs were present on the 3’UTR, and 1 clearly non-LOI SNP in the first exon. This has been added to the revised version of the manuscript.

Reviewer #2 (Remarks to the Author):

In this manuscript Goovaerts and colleagues describe the characterization of imprinting using RNA-seq data for breast cancers from the TCGA consortium. This is an extremely timely as there is currently much interest in genomic imprinting in cancer. Furthermore, they describe an informatics method, based on deviation of Hardy-Weinberg equilibrium, in that heterozygosity/biallelic SNPs will not be detected following RNA-seq, which could be applicable to other datasets without accompanying genotypes. The manuscript is well-written, easy to read and follow. It is certainly an interesting study, which after addressing all comments below will be worthy of publication and I am sure will be of interest to the readers of Nature Communications.

We deeply appreciate the reviewer’s remarks.

- The introduction is easy to read and informative for the subject. The authors correctly discuss the issues raised by recent LOI studies, i.e. LOI associated with down-regulation/no change in expression despite the prediction that there should be a doubling, and the down-regulation despite normal methylation. To emphasis that expression levels and LOI could be important in cancer biology, and that relying solely on methylation profiling maybe misleading (as a biomarker), the authors should reference the recent Nature Communication publication (doi: 10.1038/s41467-017-00639-9).

We agree with the reviewer. We have included the relevance of copy number variation (and the reference) in the Introduction section. Moreover, we explored CNV (next to DNA methylation and

the putative imprinted gene network) in breast cancer, demonstrating this to be major determinant of imprinted gene expression downregulation in cancer. This information has been added to the manuscript (Results section and Supplementary Results, Section 6 and 7).

- For example, recent studies identified... 7. This reference is not recent since it was published in 2005! Please rephrase the sentence or use a more recent publication.

This paper is indeed not recent and we have corrected this mistake.

- The authors describe a method for profiling monoallelic expression of imprinted genes and their deregulation in cancer without the need for genotype data. The rationale is that a perfectly imprinted gene will be monoallelically expressed and no heterozygous cases will be observed in RNA-seq, this resembling a homozygous sample. But how do you discriminate homozygous loci from monoallelically expressed genes, or is in fact the data a mixture of the two, with imprinting inferred due to the lack of heterozygous/biallelic genes? Please state this clearly.

At the RNA-seq level, non-imprinted loci are observed as a mixture of homozygous and heterozygous samples, with proportions approximated by the Hardy-Weinberg theorem. So for a specific locus, homozygotes as well as heterozygotes occur. If a locus is imprinted, on the other hand, only one allele is expressed and hence no heterozygotes can be observed in RNA-seq data. So, it is indeed by screening for the lack of heterozygous samples that we detect imprinting (which is also observed on the mixture distribution plots), and these samples cannot be discriminated from homozygous ones. At the gene level, we can only detect imprinted loci when there are polymorphisms present (at sufficiently high frequencies), implying that there will be heterozygous individuals at the DNA level. We have further clarified this in the text, by mentioning (i) that SNPs are required to discriminate between alleles (very beginning of Results section), (ii) that homozygous samples and heterozygous cannot be discriminated in case of 100% imprinting (in the Methods section, was already stated in the Results section).

- Of the 127 SNPs identified a possibly imprinted, mapping to 35 genes, the authors mention that 16 are already known as imprinted. However the list in Table 1 is inaccurate since SNHG14 and PWAR6 are known to be paternally expressed (also known as the UBE3A-ATS non-coding transcript from which IPW, PWARs and snoRNA and miRNAs are processed)(DOI: 10.1093/hmg/dds130). Therefore 27 SNPs should be reassigned as “known imprinted” and the numbers altered throughout the text.

We have updated this part of the text: as we have performed additional validation (cf. request infra), we do not longer compare our list with the state of the art in a quantitative manner to support the general relevance of our results, as it is rather likely that some “known” ones (also in other species) will be missed while screening literature (cf. the indeed correct examples provided by the reviewer). This part has been transferred to supplementary information (Supplementary Results, Section 3a), and SNHG14 & PWAR6 are mentioned as known imprinted loci.

- Of the list of putative imprinted genes, at least three of them are pseudogenes (MTCO1P12, CPHL1P and MTRNR2L1). How can the authors be sure that they are detecting the pseudo-copy, as stated, or the original gene when using short-read RNA-seq??

We agree with the reviewer that this may compromise our results. We hence performed a multiple sequence alignment of all reads in the BAM files covering the SNPs detected as “imprinted”. The consensus sequence was then blasted against the human reference genome to evaluate if we were detecting the pseudogene or the original gene (Supplementary Results, Section 3e). For most genes, the consensus sequences could only be aligned to the pseudogene, not the reference gene (CPHL1P, ZNF300P1, USP32P2) (note that CPHL1P could not be validated in GTE_x). For MTCO1P12 and MTRNR2L1, however, the consensus sequence aligned well to the original gene as well (even better for MTRNR2L1), yet the presence of heterozygous samples (limited partial imprinting of both loci) and clearly different allele frequencies over SNPs (for MTRNR2L1), among others, obviously contradicts a mitochondrial origin (Supplementary Results, Section 3e). Therefore, in the revised version, we have listed these loci as “candidate imprinted” in Table 1, and indicated their possible mitochondrial origin in the legend.

- The authors describe the difficulty in assigning monoallelic SNPs to overlapping transcripts and mention the MCTS2P/HM13 locus. This is true since the MCTSP2 retrogene can splice into the last eight exons of HM13. Please add a reference for this statement (DOI: 10.1093/nar/gkq1230 or alike).

We agree that this reference should have been added for our statement, the omission has been corrected.

- The imprinting (and evolutionary lack of) is very complicated and the HM13/MCTS2P locus is the basis of the imprinting by alternative polyadenylation. The description of this locus (and ideally a Supplementary Figure showing isoforms, SNPs and imprinted/LOI transcripts) should be given (DOI: 10.1101/gad.473408 and doi: 10.1093/nar/gkq1230).

A gene structure figure (Supplementary Figure 25) with information on imprinted transcripts per subtype and LOI transcript was created and added to the Supplementary Figures to complement Table 2.

- Since many reading this work may not be familiar with the bioinformatics approach, an example of a biallelically-expressed gene should be included in Figure 1. When the authors discuss partial imprinting of IGF2 in reference to Figure 1, could they give %-estimated range (i.e. 95% from one allele)?

We added a biallelically expressed gene as well as the %-estimate to the figure. For IGF2, we observed 97% imprinting.

- How can “intronic” SNPs be used to assess allelic expression? Are they detecting the primarily, unspliced transcripts?

Yes, these are probably detected from unspliced transcripts. It is important to note that the expression of these intronic SNPs was mostly lower than other SNPs, and that typically no overlapping transcripts have been reported for the introns here discussed. Some exceptions are

presented below (see also Supplementary Figures 16-17). This indicates that we indeed observe these SNPs as a result from unspliced transcripts.

- When comparing the putative monoallelic genes with TCGA WES genotypes, please indicate the number that were actually heterozygous at the DNA level and monoallelically expressed.

We have added a table with these numbers in the Supplementary Results, Section 3c (Supplementary Table 6).

- Please add a Y-axis label to all graphs.

A Y-axis has been added to all graphs.

- When discussing LOI, the authors need to be aware that MEST and HM13 have overlapping non-imprinted transcripts that may be highly expressed in the cancer sample and therefore “mimic LOI” whereas it is actually expression of a non-imprinted isoform. This promoter switching from imprinted to biallelic has been reported for MEST (PMID: 12023987 and discussed in doi: 10.1038/s41467-017-00639-9). The authors should ascertain if the samples with LOI MEST have high expression of the MEST variant_2 (NM_177524).

Our method does not allow to directly distinguish between different transcripts since detection of (loss of) imprinting is SNP-based. As also stated in our answers to reviewer 1, short-read RNA-seq data simply does not allow to perform accurate transcript level analysis.

That said, we attempted to find additional evidence that would pinpoint promoter switching. As only one significantly imprinted SNP was found for MEST, located in the 3'UTR, shared over all transcripts, no discrimination between transcripts was possible. Hence, we also evaluated the other, non-significant and/or filtered SNPs in this gene. However, only one alternative exonic SNP was available (rs1050582), which was also located in the 3'UTR shared over all transcripts and thus did not provide additional information. For your information, this SNP was originally filtered out due to a high sequencing error rate, yet demonstrated imprinting and loss of imprinting similar to the SNP reported in the manuscript (Figure 1).

Subsequently, we also evaluated exon-specific expression levels as they may also support promoter switching. We hence downloaded exon count data from firebrowse.org. Normalization was done with EdgeR and the exons were screened for differential expression between control and tumour data with a Wilcoxon Rank Sum test (same procedure as Methods, Section 7). Of the 14 exons, 12 were significantly downregulated in tumour data (and one non-significantly downregulated). The 5'UTR of the MEST variant_2 was the only exon to show slight upregulation, but was not significant. Biallelic expression of the MEST variant_2 and downregulation of the imprinted transcripts in cancer may thus (partly) cause the observed “loss of imprinting” in the latter. Moreover, it may also explain the limited “loss of imprinting” observed in controls, cf. Figure 2(a) in the manuscript. However, we cannot confirm this with the available data. The same analysis was done for HM13, here no differences were found between exons and thus between the varying transcripts. This was described in the Results section of the manuscript. Moreover, the relevance of alternative transcripts is more stressed in the revised version of the Discussion section.

Figure 17 Mixture distributions of rs1050582 located in MEG3. The SNP was filtered due to high sequencing error rate, but shows clear imprinting in (a) control samples and loss of imprinting in (b) tumour samples.

- The non-imprinted isoform of MEST is tissue-specific and expressed in blood only (DOI: 10.1086/302712). Do the authors see any correlation between infiltrating lymphocytes and the presence of MEST biallelic expression?

As information on infiltrating lymphocytes is available for TCGA data, we checked the tumour data for significant correlation between biallelic expression (we used the allelic ratio as an estimate for biallelic expression) and number of infiltrating lymphocytes. No significant correlation (with a permutation correlation test which only takes the putative heterozygous samples into accounts, cf. detection of LOI, p-value = 0.9) was found for the MEST locus. Furthermore, MEST was downregulated in tumour samples, suggesting that infiltrating lymphocytes are not the explanation of biallelic MEST expression (Supplementary Results, Section 8).

Also HM13 is of interest, and we hence performed the same analysis for the imprinted HM13 SNPs. Only rs1115713, featured by imprinting but no LOI, was detected with a significant p-value (p-value of 0.01 and $R = 0.15$, see Supplementary Results, Section 8). Note that this SNP is exonic in MCTS2P and that the correlation could be rather due to biallelic MCTS2P expression than because of HM13 expression.

- The authors need to clarify whether the inconsistency described for SNPs within the same gene is actually due to technical/power associated causes as stated, or due to re-expression of overlapping non-imprinted isoforms as could occur for a large number of imprinted genes (PLAGL1, GRB10, MEST, MCTS2/HM13, INPP5F/INPP5Fv2, NAPIL5/HERC3, PPIEL, ERLIN2, ZNF331, NAA60, WRB and SNU13).

We analysed all (significant and non-significant) SNPs in the stated genes. Most genes for which we did not observe any imprinting (GRB10, INPP5F, ERLIN2, NAA60, WRB and SNU13, Figure 4, 6, 7, 9, 10) were clearly not imprinted independent of overlapping transcripts. Only PPIEL was missed due to technical issues, mainly low coverage for putatively imprinted SNPs. Note that this was also the locus found by Baran et al. that we missed, we have added this to the Discussion section. Figure 2 illustrates that overlapping transcripts may disguise imprinting, yet the evidence/coverage was too poor to call this an imprinting candidate in our data. For MEST and

NAP1L5 imprinting was clear in the actual genes (non-significant SNPs only due to technical issues, Figure 8 and 3), whereas the overlapping genes were certainly not imprinted. Only ZNF331 showed signs of transcript-specific imprinting (Figure 8 and Supplementary Figure 16(c)), which is also represented in the more complex imprinting patterns (see mixture distributions, Figure 7)

Figure 2 Gene structure map of PPIEL and a mixture distribution plot of a non-significant SNP (indicated with a red arrow on the transcript map) located in PPIEL/BMP8A. More complex patterns of imprinting are observed here which could be explained by overlapping transcripts. On the transcript map SNPs that were filtered for technical reasons are marked with a #, filtered imprinted SNPs that could be imprinted with * and non-imprinted SNP with a °.

Figure 3 Gene structure map of NAP1L5 and HERC3 with two mixture distribution plots: an imprinted SNP located in NAP1L5 and a non-imprinted one located in HERC3 (indicated with a red arrow on the transcript map). On the transcript map significantly imprinted SNPs are marked with an “i” and non-imprinted ones with °.

Figure 4 Gene structure map of GRB10 and a mixture distribution plot indicating that this genes was clearly not imprinted (indicated with a red arrow on the transcript map). On the transcript map non-imprinted SNPs are marked ° and SNPs that were filtered for technical reasons with a #.

Figure 5 Gene structure map of MEST/MESTIT1 and a mixture distribution plot of SNP that looks (partially) imprinted, but was missed due to technical issues (indicated with a red arrow on the transcript map). On the transcript map significantly imprinted SNPs are marked with an “i”, filtered imprinted SNPs with * and SNPs that were filtered for technical reasons with a #.

Figure 6 Gene structure map of INPP5F and a mixture distribution plot indicating that this gene was clearly not imprinted (indicated with a red arrow on the transcript map). On the transcript map non-imprinted SNPs are marked ° and SNPs that were filtered for technical reasons with a #.

Figure 7 Gene structure map of ZNF597/NAA60/MIR6126 and a mixture distribution plot of a SNPs that is clearly not imprinted (indicated with a red arrow on the transcript map). The transcript map shows that indeed only ZNF597 is imprinted. On the transcript map significantly imprinted SNPs are marked with an “i”, filtered imprinted SNPs with * and non-imprinted ones with °

Figure 8 Gene structure map of ZNF331 and a mixture distribution plot of a non-imprinted SNP located in ZNF331 (indicated with a red arrow on the transcript map). More complex patterns of imprinting are observed here which could be explained by transcript-specific imprinting. On the transcript map significantly imprinted SNPs are marked with an “i” and non-imprinted ones with °.

Figure 9 Gene structure map of WRB and a mixture distribution plot indicating that this genes was clearly not imprinted (indicated with a red arrow on the transcript map). On the transcript map non-imprinted SNPs are marked ° and SNPs that were filtered for technical reasons with an #.

Figure 10 Gene structure map of SNU13 and a mixture distribution plot indicating that this genes was clearly not imprinted (indicated with a red arrow on the transcript map). On the transcript map non-imprinted SNPs are marked ° and SNPs that were filtered for technical reasons with an #.

Figure 11 Gene structure map of C22orf46 and a mixture distribution plot indicating that this genes was clearly not imprinted. On the transcript map non-imprinted SNPs are marked °.

- For the LOI observed in HER2 and BL subtypes the number of individual samples showing biallelic expression for each of the 8 genes should be given.

We have tried to identify the number of LOI samples for the significant loci using whole exome sequencing data. By determining the heterozygous samples on the RNA as well as the DNA level, we would have been able to strictly call LOI samples. However, as stated in the manuscript, low coverage hampered genotype calling in whole exome sequencing data. We therefore decided to genotype the samples with SeqEM (assuming HWE) on the RNA-seq data. These results give an indication of how many samples show biallelic expression and hence lost their imprinting signature. However, it should be noted that we cannot detect LOI in homozygous samples, so we used HWE to provide a more accurate estimation of the fraction of LOI samples. Supplementary Table 8 (Supplementary Results, Section 4b) was added with these results.

- For the LOI at the H19/IGF2 locus, was biallelic H19 expression associated with higher levels of IGF2?

In tumour data, biallelic expression of H19 was significantly correlated with higher expression of IGF2 for 4 of the 10 SNPs detected as imprinted in H19 in normal tissue (with a correlation permutation test for the putatively heterozygous samples, cf. detection of LOI). Interestingly, these 4 significant SNPs are all located in the same exon of transcript 1 and 2, while the other SNPs are from different exons (shared over all transcripts). As for MEST and HM13, this could indicate that different transcripts and/or promoter switching play a role in LOI, but we could not find sufficient evidence to prove this nor could we provide readers with a straightforward interpretation. These results are therefore not shown in the revised version of the manuscript, yet as mentioned higher, the putative roles of promoter switching & different transcripts (also in LOI) are more clearly discussed in the Discussion section.

- The section describing down-regulation, rather than over-expression, is interesting and important. However, the section would be more informative if the number of under/over-expressing tumour samples per subtype were given in a supplementary table (or the data presented in a figure), the range of this aberrant expression and if it was associated with genomic deletions/amplifications.

We have added Supplementary Figure 22 showing which samples exhibit over- or underexpression for specific loci and whether this is linked with gains or losses. This figure clearly shows that copy number variations and over-/underexpression is abundant in breast cancer compared to healthy breast samples. Furthermore, we performed a linear regression model to link CNV with DE. As described in Supplementary Results, Section 6, DE was frequently associated with CNV (Supplementary Table 18 and 19), with the clearest example of upregulation linked to gains in HM13 and many downregulated genes associated with losses. However, deregulated expression of imprinted genes in breast cancer is not always explained by CNV, for example no significant results were found for MEG3 and DLK1 (Supplementary Table 18).

As for HM13 many LOI samples were found, we were also interested in the association between LOI and CNV for all imprinted HM13 SNPs. A χ^2 test showed that LOI occurred independently of CNV for all loci (Supplementary Table 20).

- LOI is a very well-defined term. As mentioned in the introduction it is the re-expression of the normally silent allele. Therefore is it possible to introduce a new term for the interesting observation that biallelic expression was associated with down-regulation of the normally expressed allele so that both alleles are more equally expressed at a residual level and perceived as LOI?

To avoid confusion, it is definitely a good idea to use a different term than LOI. We called the observation of down-regulation linked to biallelic expression silencing induced LOI (siLOI).

- Why do the authors use different FDR cut-offs in Table 2 for LOI (≤ 0.1) and DE (≤ 0.05)? Surely it would be best to use 0.05 throughout, which would then highlight that in a heterogeneous group of breast tumours LOI may not be significant, but with the correct classification significant LOI groups are observed.

We agree that it would be better to use the same FDR cut-off throughout the whole manuscript. We evaluated the option suggested, yet it became clear that some loci featured by LOI would disappear from the analysis. E.g. ZNF331 shows LOI (with FDR <0.1 but >0.05) for two SNPs in two tumor subtypes (Table 2 in main text), but is irrelevant in the other tumor types. Though sufficient evidence for LOI (in our opinion), this locus would need to be removed from the LOI results list.

Moreover (and the original reason why two different FDR cut-offs were used), we have two types of analyses with major differences in sensitivity (detection of imprinting vs. detection of LOI). Indeed, the power decreases heavily for the LOI analyses as homozygous samples (for which no LOI can be observed) are in the majority. We hence decided to use a different FDR cut-off for the stated analyses to enable better detection of LOI, at the cost of lower specificity. As the adjusted FDR values are provided, this can be easily interpreted by the readers. This rationale is also described in the revised Results section of the manuscript. Note that reducing specificity/increasing sensitivity for imprinting detection is not necessary, as sensitivity is sufficiently high.

- When comparing the HM450k methylation data for MCTS2P DMR within the HM13 gene, the authors only describe correlations for individual probes and increased expression. Is this because the (average) methylation over the entire DMR did not show any association? The MCTS2P DMR contains 10 probes (doi: 10.1080/15592294.2016.1264561 and doi: 10.1038/s41467-017-00639-9) and a supplementary figure should be included showing the methylation for all MCTS2P DMR probes not just those with significant correlations.

The original reason for not using average methylation are the locus (and thus probe) specific effects of methylation, with “typically” e.g. a negative association between expression and promoter methylation and a positive association for exon methylation, yet with many exceptions. For example, when using MEXPRESS (mexpress.be) to plot the association between HM13 expression and probe-specific methylation, we observe many differences in methylation of the varying probes (see also Supplementary Figure 25). Nevertheless, one would indeed anticipate that all probes within the MCTS2P DMR behave similarly, and we have added the average value to the Supplementary Table 22-23. Moreover, we have now included information for all individual probes in Supplementary Figure 26, yet only analysed probes that showed hemimethylation (which were but one located in the DMR), as clear from this figure. Note that for one probe (cg14175568), no data were available in TCGA (also for other tumours), probably due to low quality probe results (typically interference with a SNP or probe intensities not significantly higher than background).

- The authors describe several genes for which imprinting has not been confirmed (MTCO1P12, LINC01139, PAX8-AS1, PTX3, CPHL1P, ATP8A1, ZNF300P1, LOC100294145, HOTAIRM1, RP11-109L13.1, GLIPR1/KRR1, PLIN1, TPSB2, USP32P2, MTRNR2L1, BCR). Rather than adding to the list of conflicting genes, which often hamper and confuse subsequent studies, the authors should present data confirming allelic expression (not only in breast but any tissue) or germline methylation.

The authors agree that providing novel genes without additional confirmation may lead to more confusion in the field. On the other hand, tissue specific imprinting, the distinction between

primary and secondary imprints, the often complicated link between DNA methylation and expression (cf. IGF2/H19 locus) and the presence of alternative regulatory mechanisms (such as histone modifications) does complicate validation with other tissues or with DNA methylation. Therefore, the authors have done a major effort to evaluate all imprinted loci in GTEx breast samples: upon approval by dbGAP, GTEx breast RNA-seq data from 92 samples were downloaded, pre-processed and screened for imprinting in the set of imprinted genes detected in TCGA. Loci with a median imprinting level below 0.4 were also filtered out. Validation was possible for roughly 97% of all SNPs, and 86% of all genes (typically only a single SNP was detected in TCGA for non-validated loci). This led to the elimination of CPHL1P, ATP8A1, GLIPR1/KRR1 and TPSB2 from our set of imprinted genes. Four genes showed either a low allele frequency or low coverage, making validation difficult. These genes are denoted as “candidate imprinted”, namely PTX3, RP11-109L13.1, PLIN1 and USP32P2. MTRNR2L1 was not found in GTEx data, but was clearly imprinted in TCGA and was hence not eliminated. However, it should be noted that the latter gene is currently mentioned as “candidate imprinted” given that we could not fully exclude a mitochondrial origin (see also higher).

- No evidence for an imprinted breast-cancer IGN has been presented. Minimal evidence should be given, for example do deregulated samples show LOI for multiple genes? Otherwise please refrain from mentioning it in the abstract discussion.

We have added correlation and principal component analyses to substantiate our claim of an imprinted breast-cancer IGN (Supplementary Results, Section 6 and 7). Interestingly, we identified PLAGL1 (cf. prostate cancer IGN), MEG3 and ZNF300P1 as possible regulators. Given that MEG3 CNV was not associated with expression in cancer (and that this association was rather limited for PLAGL1 and ZNF300P1), evidence supports the presence of an IGN in breast cancer independent of CNV.

Reviewers' comments:

Reviewer #2 (Remarks to the Author):

The authors have made a good effort to address my comments. This is a great paper which I am sure will be referenced often. I have just a few minor comments that need addressing:

- The authors should include the isoform-specific maps in reviewers comments to suppl info as this highlights the pitfalls of short-read RNA-seq for imprinting.
- I am still not sure that silence induced loss-of-imprinting is the correct term. I think is it much more accurate to use something like residual biallelic expression (RBE)??
- A hemimethylated profile (this is an in correct term for allelic methylation as it is specific to DNA strands during replications) is not particularly informative for a DMR, especially if restricted to a single probe (since most imprinted DMRs are 300bp-1 kb in size). For the "additional validation step" of candidate imprinted genes the authors should interrogate the intervals using the genome-wide uniparental diploidy samples used to characterised parent-of-origin methylation (doi: 10.1101/gr.164913.113).

The authors would like to thank Reviewer #2 for his valuable feedback and constructive criticisms which have undoubtedly increased the overall depth and quality of the manuscript. Next to modifications addressing the reviewer's comments, we have also made other minor modifications to the manuscript, predominantly to adhere to Nature Communication's detailed guidelines, including the abstract and manuscript word limits, several style aspects as well as essential technical information, e.g. with respect to data and code availability (Methods, Section 9 and 10). Other minor modifications include correction of a few remaining typographic errors, addition of secondary affiliation for some authors, and reference to GTEx and TCGA data in line with the guidelines of the corresponding consortia.

Reviewer #1 (Remarks to the Author):

As you know, reviewer #1 was unable to comment further at this time. Reviewer #2 kindly agreed to assess your response to reviewer #1's previous comments. Reviewer #2 agreed with reviewer #1 that the authors should not refer to transcriptional networks (in abstract or manuscript) without performing appropriate co-expression analyses, but felt the other issues raised by reviewer #1 had been adequately addressed by your response.

The authors are thankful for Reviewer #2's willingness to address these comments as well. The authors agree that co-expression analysis is required to formally identify imprinted gene networks. We do not longer refer to imprinted gene networks in the abstract and with respect to our own results, except for the discussion section where we argue that our findings are compatible with (rather than proof of) the presence of such networks (Supplementary Note 7).

Reviewer #2 (Remarks to the Author):

The authors have made a good effort to address my comments. This is a great paper which I am sure will be referenced often. I have just a few minor comments that need addressing:

- The authors should include the isoform-specific maps in reviewer's comments to suppl info as this highlights the pitfalls of short-read RNA-seq for imprinting.

We have added these maps to the Supplementary Information section (Supplementary Fig. 18-19), where we also mention the limitation of using short-read RNA-seq data for this purpose.

- I am still not sure that silence induced loss-of-imprinting is the correct term. I think is it much more accurate to use something like residual biallelic expression (RBE)??

The main question is whether the alleged loss-of-imprinting (LOI) associated with imprinted gene silencing (thereby revealing the presence of residual biallelic expression) should be formally considered LOI or not. Though also coined LOI by other groups, we agree that – for the purpose of clarity – it would be better to use an alternative term, underscoring the different underlying biology. In the manuscript, the term “differential imprinting” was already used in several instances, and we have therefore replaced LOI with “differential imprinting” to indicate increased *relative* biallelic expression in general (Results, section 2 and Methods, section 5). Only for those cases where the silenced allele is re-expressed in cancer (absolute higher expression of the silenced allele, not only relative), we now use the term LOI (Results, section 3 and 5). Though we have considered to use the term “RBE”, this was complicated by the fact that RBE provides an explanation for the “perceived” LOI, but does not refer to the latter directly. Moreover, we cannot exclude that RBE is also present in loci genuinely featured by LOI in cancer, though it will of course

not be the cause of the LOI. We have adapted the manuscript accordingly, and have included a brief rationale concerning the terminology in the discussion section.

- A hemimethylated profile (this is an incorrect term for allelic methylation as it is specific to DNA strands during replications) is not particularly informative for a DMR, especially if restricted to a single probe (since most imprinted DMRs are 300bp-1 kb in size). For the “additional validation step” of candidate imprinted genes the authors should interrogate the intervals using the genome-wide uniparental diploidy samples used to characterise parent-of-origin methylation (doi: 10.1101/gr.164913.113).

The term hemimethylation indeed refers to strand-specific methylation and has been replaced with “monoallelic methylation” throughout the manuscript (Results, section 1 and 4). We agree with the reviewer that the results described in the latter publication (Court *et al.*, Genome Res 2014) should be considered the gold standard for individual parental allele marking differentially methylated regions (DMRs). However, Court *et al.* integrated heterogeneous data using a complex methodology and several data filters, making it impossible for us to construct a clear background of non-DMRs, which is essential to statistically evaluate enrichment in our set of candidate imprinted loci.

Vice versa, the significant enrichment observed in our analysis does not prove DMR status of individual loci, but suggests that DMRs are present far more frequently in our set of imprinted loci than randomly expected. As neither approach can fully address the question at hand, we opted to integrate them both in the results section of the revised manuscript. First, we mention the result of the (original) enrichment analysis, but continue by pointing out that allelic methylation is only a crude proxy (cf. the reviewer’s comment). Subsequently, we describe (qualitatively) how the DMRs identified by Court *et al.* (Table 1 of the latter publication) overlap with our set of imprinted genes, and discuss the presence of (putative) allelic and differentially methylated probes (cancer – control) within these DMRs (Results, section 1 and 4). The DMR annotation information has also been added to the methylation data of Supplementary Table 7.

REVIEWERS' COMMENTS:

Reviewer #2 (Remarks to the Author):

The authors have now answered all queries. I only suggest two changes to the manuscript:

Line 87: "the increased frequency of biallelic expression is far more...."

Line248: remove "though the imprinting status itself remains unaffected".

Reviewer #2 (Remarks to the Author):

The authors have now answered all queries. I only suggest two changes to the manuscript:

Line 87: "the increased frequency of biallelic expression is far more...."

Line248: remove "though the imprinting status itself remains unaffected".

The two sentences have been adjusted in the manuscript.